# Signature-driven repurposing of Midostaurin for combination with MEK1/2 and KRASG12C inhibitors in lung cancer

Drug combinations are key to circumvent resistance mechanisms compromising response to single anti-cancer targeted therapies. The implementation of combinatorial approaches involving MEK1/2 or KRASG12C inhibitors in the context of KRAS-mutated lung cancers focuses fundamentally on targeting KRAS proximal activators or effectors. However, the antitumor effect is highly determined by compensatory mechanisms arising in defined cell types or tumor subgroups. A potential strategy to find drug combinations targeting a larger fraction of KRAS-mutated lung cancers may capitalize on the common, distal gene expression output elicited by oncogenic KRAS. By integrating a signature-driven drug repurposing approach with a pairwise pharmacological screen, here we show synergistic drug combinations consisting of multi-tyrosine kinase PKC inhibitors together with MEK1/2 or KRASG12C inhibitors. Such combinations elicit a cytotoxic response in both in vitro and in vivo models, which in part involves inhibition of the PKC inhibitor target AURKB. Proteome profiling links dysregulation of MYC expression to the effect of both PKC inhibitor-based drug combinations. Furthermore, MYC overexpression appears as a resistance mechanism to MEK1/2 and KRASG12C inhibitors. Our study provides a rational framework for selecting drugs entering combinatorial strategies and unveils MEK1/2- and KRASG12C-based therapies for lung cancer.

Targeted therapies often display limited antitumor effect due to intrinsic and adaptive resistance mechanisms, underscoring the need for rational combinatorial treatments to yield better and more durable responses[1,2]. Given the relevance of the RAF-MEK-ERK pathway in KRAS-driven lung cancer[3], particularly in the most frequently diagnosed subtype, lung adenocarcinoma (LUAD), combinations of MEK1/2 inhibitors (MEKi) with targeted agents to proximal elements of the KRAS signaling network have been reported[4–12]. Likewise, recent studies using KRASG12C inhibitors (KRASG12Ci) have illustrated the requirement of additional pathway inactivation for deep antitumor responses[13–19]. Overall, drugs selected for dual strategies have been nominated by previous knowledge on proximal signaling pathways involved in KRAS oncogenesis or in adaptive resistance to

MEKi. However, the efficacy of such drug combinations is generally restricted to subsets of mutant (mut) *KRAS* LUAD with specific signaling pathway activation/reactivation or epithelial/mesenchymal phenotypes[7,12,20]. An alternative to nominate drugs entering combinatorial approaches whose effect spans a large fraction of patients may capitalize on the common gene expression output originated by KRAS oncoprotein that is featured in LUAD, but is yet to be investigated.

Drug repurposing is an effective strategy to find new indications for existing drugs already tested for safety, dosage, and toxicity[21]. The development of combinatorial approaches that could rapidly progress to the clinic may be expedited by the identification of repurposed drugs. One drug repositioning approach utilizes transcriptional

e-mail: silvevicent@unav.es

profiles generated from cell lines exposed to a large compendium of pharmacological compounds[22] and aims to predict drugs that reverse the expression of a gene signature representative of a certain disease[23]. A successful example of this approach in lung cancer was the identification of tricyclic antidepressants as potential therapeutics for small-cell lung cancer[24]. However, such drug repurposing strategy has so far not been fully exploited for the development of therapies in lung cancer. Moreover, neither has it been implemented in the context of specific oncogenic drivers nor applied for the nomination of hits entering combinatorial approaches.

Here, through the development of a drug repurposing approach coupled to a pairwise screen, we identify the combination of MEKi (Trametinib) with multityrosine kinase PKC inhibitors (mtPKCi; Lestaurtinib and Midostaurin) as an effective therapeutic strategy for a large percentage of mut *KRAS* LUAD in vitro and in vivo. We also show that the KRASG12Ci Sotorasib can substitute for Trametinib with similar efficacy. Additionally, we provide mechanistic evidence related to the consequences of the dual treatments through global and focused protein analyses, as well as its relationship to clinical outcome. Overall, our study reports therapeutic strategies involving drugs already approved by the FDA for the treatment of a large spectrum of *KRAS*-mutated LUAD.

## Results

### Synergistic drug combinations in mut *KRAS* LUAD unveiled through a drug repurposing-based strategy

To identify drugs with potential activity against mut *KRAS* LUAD, we performed a gene expression-based drug repurposing approach using the extended version of an interspecies *KRAS* gene signature (iKRASsig)[25]. IKRASsig upregulated genes were enriched in mut *KRAS* LUAD patients compared to wild type (wt) individuals in several data sets[26–31] (Suppl. Fig. 1A). Moreover, high expression of iKRASsig upregulated genes along with mutations in *KRAS* identified LUAD patients with the worst survival outcome (Suppl. Fig. 1B). Given the tight association of the iKRASsig with *KRAS* genotype, we queried the Connectivity Map[32] to find drugs that potentially reverse the gene expression signature (Fig. 1A). We posited that drugs predicted to have the lowest repurposing score (RS) (i.e., more effectively reverse the iKRASsig) would a priori have an adverse effect on mut *KRAS* tumors. Several predicted drugs were identified (RS < −0.3), including inhibitors against KRAS effectors, what supported the genotype specificity of the repurposing approach (Fig. 1B and Suppl. Table 1).

The predicted drugs or alternative compounds against the same targets (Fig. 1B and Suppl. Table 1) were then screened in a pairwise format in two mut *KRAS* LUAD cell lines (H1792: *KRAS*G12C; H2009: *KRAS*G12A) using concentrations equal or lower than their IC50. Compusyn analysis unveiled several synergistic combinations in at least one cell line (combination index, CI < 0.8) (Fig. 1C and Suppl. Fig. 1C). One of them involved a MEKi (Trametinib) and a pan-HERi (Neratinib) combination reported to have an adverse effect on a subset of mut *KRAS* LUAD[7,12]. Two drug combinations, corresponding to those of Trametinib with the multityrosine kinase PKCi (mtPKCi) Lestaurtinib (also regarded as FLT3i), and Trametinib plus the WEE1i Adavosertib, were synergistic and had a large antiproliferative effect compared to single drugs in the two cell lines (Fig. 1D, E). We selected the combination consisting of drugs in the latest clinical stages: Trametinib and Lestaurtinib. Trametinib is already approved for the treatment of *BRAF*-mutated melanoma and lung cancer in combination with Dabrafenib[33,34], and Lestaurtinib, a staurosporine derivative, completed a phase III clinical trial for the treatment of acute myeloid leukemia (AML)[35].

To start understanding the synergistic effect of Trametinib and Lestaurtinib, we performed RNAseq analysis on H1792 cells treated with single drugs (Suppl. Data 1). Principal component analysis (PCA) showed distinct clustering of the samples treated with individual drugs (Fig. 1F). In addition, both drugs elicited highly specific gene changes (Suppl. Fig. 1D, E). Furthermore, Trametinib and Lestaurtinib reversed the expression of different gene clusters within the iKRASsig (Fig. 1G). These observations support the synergistic effect of the dual treatment on *KRAS* mutation-bearing LUAD cells.

### Preferential sensitivity of mutant *KRAS* LUAD cells to dual Trametinib and Lestaurtinib treatment in vitro and in vivo

Since the two cell lines used for the pharmacological screen had KRAS mutations, we asked whether the antitumor effect was dependent on the *KRAS* genotype by using four additional mut and five wt *KRAS* LUAD cell lines. The drug combination affected all mut *KRAS* cell lines more extensively than single treatments, while no such pattern was seen in wt *KRAS* ones (Fig. 2A and Suppl. Fig. 2A).

Using an adapted protocol for the growth of tumor organoids[36,37], we also tested the drug combination under 3D conditions, since mut *KRAS* lung cancer cells display a higher dependence on KRAS oncogene signaling in 3D cultures than 2D counterparts[38,39]. The dual treatment had a larger antiproliferative effect than single drugs (Fig. 2B).

We then tested the efficacy of the drug combination in vivo. To do this, cell line-derived xenografts (CDXs) from H1792 cells were generated and exposed to single and double treatments when tumor volume reached ~100 mm³. Mice treated with both Trametinib and Lestaurtinib had smaller tumors than those mice treated with single drugs (Fig. 2C, D). Of note, the antitumor effect ran without changes in mouse weight (Suppl. Fig. 2B). Interrogation of the cellular mechanism behind the in vivo antitumor effect showed a robust induction of apoptosis by the drug combination, as measured by cleaved PARP or Annexin V expression, in mut *KRAS* LUAD cells but not in wt ones (Fig. 2E, F and Suppl. Fig. 2C). These results suggest that the effect of the drug combination is mainly cytotoxic and genotype specific, and provide the proof-of-principle on the use of mtPKC plus MEK1/2 inhibitors as a potential therapy for mut *KRAS* LUAD.

### The FDA-approved Lestaurtinib analog Midostaurin synergizes with Trametinib

To increase the translational value of these findings, we searched for drugs in clinical use structurally related to Lestaurtinib that could yield a similar effect in combination with Trametinib. We chose another staurosporine derivative mtPKCi, Midostaurin (PKC412), approved for AML and advanced mastocytosis treatment[40,41]. Trametinib and Midostaurin combination was also tested in mut and wt *KRAS* LUAD cell lines, including an additional cell line isolated from a *KRAS*-mutated LUAD patient (CP435). The dual treatment induced a larger antiproliferative effect than single drugs exclusively in cell lines with *KRAS* mutations in 72-hour treatment assays (Fig. 3A, B, and Suppl. Fig. 3A). This effect was further confirmed in cells treated for 5 days, where lower concentration of the two drugs were required to yield a similar effect (Suppl. Fig. 3B). A detrimental effect was also found in LUAD cell lines from preclinically relevant genetically-engineered mice bearing the most frequent human *KRAS* mutations (G12D, G12C and G12V)[42,43] (Suppl. Fig. 3C). Studies in 3D cultures revealed that the drug combination had a larger negative impact in organoid proliferation than single drugs, an effect that was further confirmed in tumor organoids from LUAD patient-derived xenografts (Fig. 3C, D).

At the molecular level, the drug combination effect coursed with apoptosis induction exclusively in human LUAD cells harbouring *KRAS* mutations (Fig. 3E, F, and Suppl. Fig. 3D, E). Interestingly, no consistent inactivation of KRAS canonical effectors or kinases involved in MEKi resistance mechanisms that would explain the cytotoxic effect was observed (Fig. 3G and Suppl. Fig. 3F). On the contrary, we found increased phosphorylation of AKT, suggesting a role as potential scape mechanism to the dual treatment.

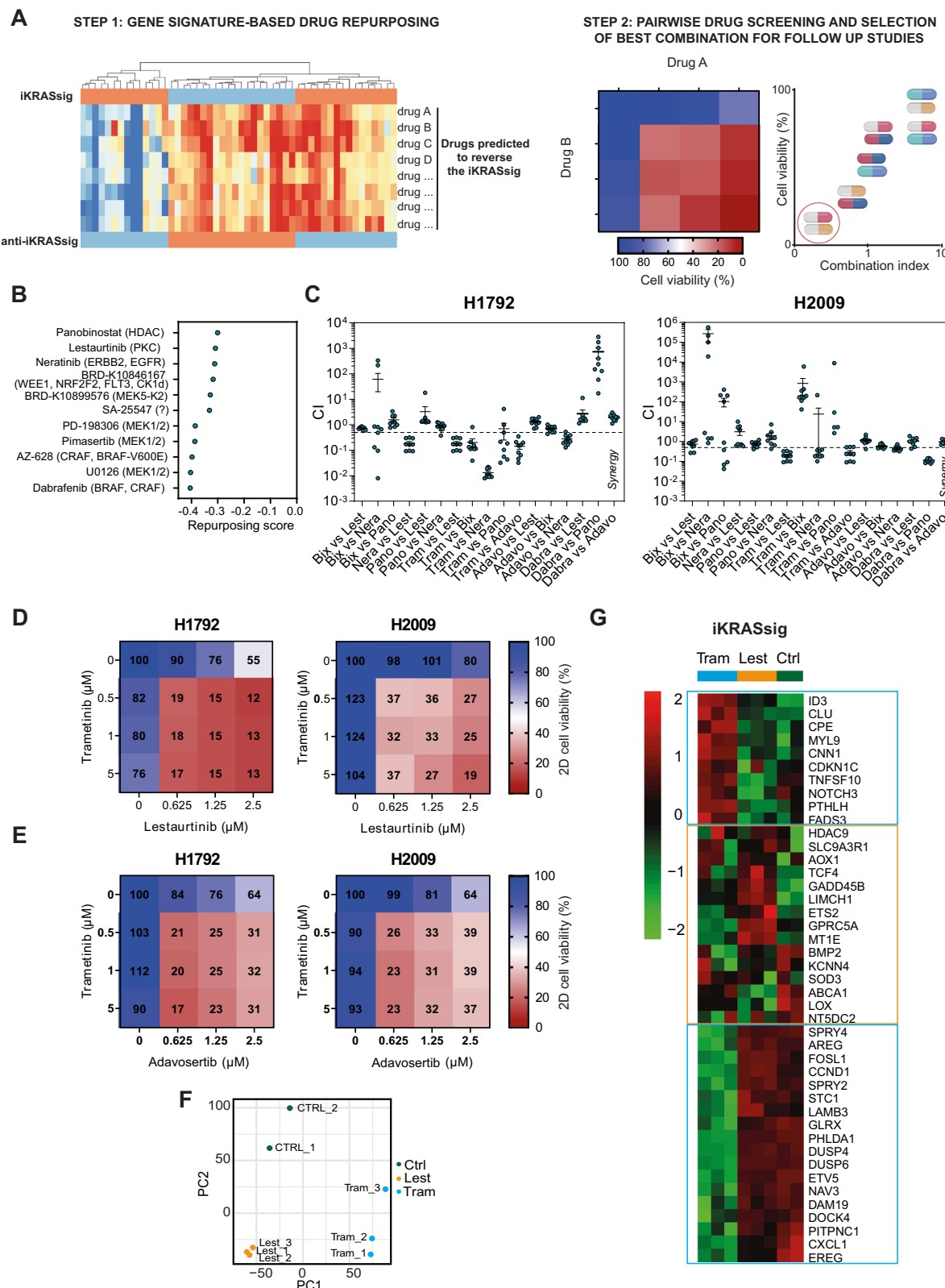

**A** STEP 1: GENE SIGNATURE-BASED DRUG REPURPOSING

STEP 2: PAIRWISE DRUG SCREENING AND SELECTION OF BEST COMBINATION FOR FOLLOW UP STUDIES

Non-genetic early resistance to MEKi compromises treatment efficacy[5]. Thus, we tested the drug combination using 10 day-treatment clonogenic assays, which would enable the development of early resistance mechanisms. The drug combination had a larger antiproliferative effect than single drugs in mut but not in wt *KRAS* cells (Fig. 3H and Suppl. Fig. 3G). We also investigated the consequences of Trametinib and Midostaurin combination in the context of long-term resistance mechanisms. Mut *KRAS* cell lines were made resistant to the

MEKi after one month of exposure to increasing concentrations of Trametinib. Trametinib-resistant (TR) cells displayed a proliferative capacity similar to parental cells (Suppl. Fig. 3H). Trametinib-resistant cell lines were not addicted to the MEKi as they displayed even higher growth kinetics than parental cells upon Trametinib removal (Suppl. Fig. 3I). This is in contrast to LUAD cells resistant to EGFRi which undergo cell death upon drug withdrawal[44], what may imply different adaptive mechanisms. Of note, TR cells remained sensitive to the dual

**Fig. 1 | Synergistic dual combinations for mutant _KRAS_ lung cancer obtained through a drug repurposing-based strategy. A** Experimental workflow employed to identify drug combinations with the highest antitumor effect on mutant (mut) _KRAS_ LUAD. **B** Repurposing scores of drugs obtained from the Connectivity Map at the Library of Integrated Network-based Cellular Signatures (LINCS) program (c3.lincscloud.org) using the interspecies _KRAS_ signature as input. Targets of predicted drugs are shown in brackets. **C** Combination index (CI) values corresponding to all concentrations for each drug-pair combination tested in mut _KRAS_ cell lines (H1792 and H2009; $n = 9$). CI < 0.8, synergism. Trametinib (Tram: MEK1/2i);

BIX02189 (BIX: MEK5-Kinase2i); Neratinib (Nera: ERRB2i, EGFRi); Lestaurtinib (Lest: PKCi), Dabrafenib (Dabra: BRAFi, CRAFi); Adavosertib (Adavo: WEE1i), Panobinostat (Pano: HDACi). Data: mean +/− SEM. **D, E** Percent cell viability of H1792 and H2009 cells treated with different concentrations of Tram and Lest (**D**) or Tram and Adavo (**E**), individually or in combination (data from the drug screening). **F** Principal component analysis (PCA) of H1792 cells treated with DMSO (Ctrl), Tram or Lest. **G** Unsupervised clustering heatmap of iKRASsig genes' expression in H1792 cells treated with DMSO (Ctrl), Tram or Lest.

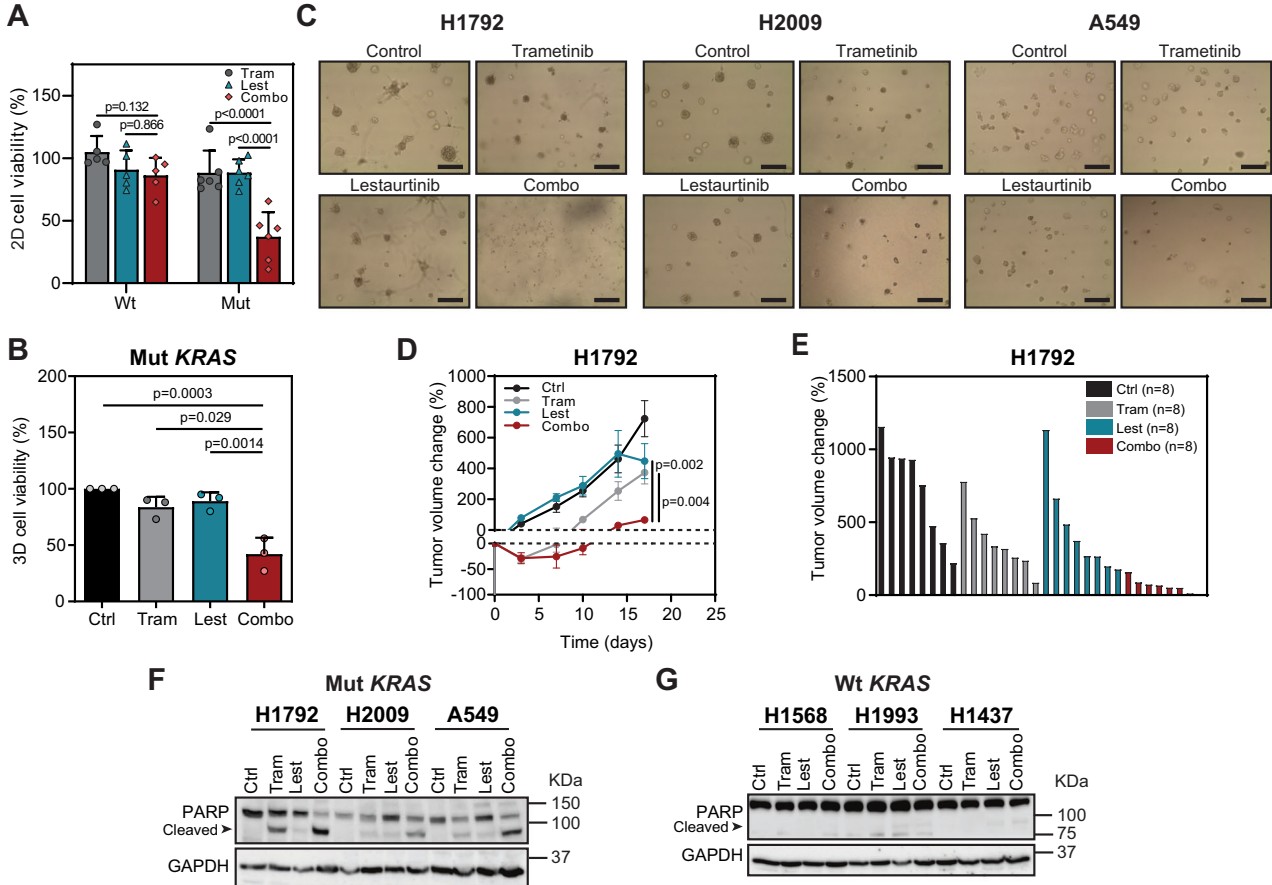

**Fig. 2 | Trametinib and Lestaurtinib combination is preferentially effective in mutant _KRAS_ compared to wild type _KRAS_ LUAD cells. A** Effect of Trametinib (Tram) and Lestaurtinib (Lest) combination on cell viability of wild type (wt) _KRAS_ (H1437, H2126, HCC78, H1993 and H1650) and mutant (mut) _KRAS_ (H1792, H2009, A549, HCC44, H23 and H358) cells treated for 72 h treatment. Tram: 0.5 μM; Lest: 0.625 μM (data: mean +/− SD; test: one-way ANOVA, Tukey's adjustment). **B** Effect of Tram and Lest combination on cell viability of mut _KRAS_ LUAD cells (H1792, H2009 and A549) grown in 3D culture conditions (72 h drug treatment; _n_: 3 cell lines). Tram: 0.01–0.05 μM; Lest: 0.05–0.1 μM (data: mean +/− SD; test: one-way

ANOVA, Tukey's adjustment). **C** Representative images of mut _KRAS_ LUAD cell lines in (**B**) exposed to the different treatments. Scale bar: 200 μm. **D** Percent fold change growth of H1792-derived tumors of mice (_n_: 8 tumors per group) treated with indicated drugs (data: mean +/− SEM; test: Kruskal–Wallis, Dunn's adjustment). **E** Waterfall plot of tumors in (**C**) at the last day of experiment. Ctrl: untreated control; Tram: 1 mg/kg; Lest: 30 mg/kg; Combo: drug combination. **F, G** Western blot of cleaved PARP expression of mut _KRAS_ (H1792, H2009 and A549; **E**) and wt _KRAS_ (H1568, H1993 and H1437; **F**) cell lines 24 h after drug treatment (loading control: GAPDH).

treatment (Fig. 3I), suggesting overlapping treatment response mechanisms with parental cells.

Multiple kinases have been reported as putative targets of the mtPKCi Midostaurin[45]. To dissect those ones primarily contributing to the synergistic effect, we took advantage of a published loss-of-function screen aimed to unveil kinases sensitizing to Trametinib in lung cancer[7]. Deep analysis of 38 putative Midostaurin kinases revealed that genetic inhibition of _PRKCH_, _PRKCA_, _AURKB_ and _MARK2_ enhanced Trametinib antiproliferative effect by more than 30%, close to those levels obtained by _FGFR1_ inhibition in the study by Manchado et al.[7]. Notably, inhibition of other kinases, such as _FLT3_, a known target in _FLT3_-mutated AML[35], had a much more residual influence (-6%) (Fig. 3J and Suppl. Fig. 3J), most likely due to low expression and/or

activation of this target in LUAD cells (Suppl. Fig. 3K). To validate the results of the genetic screen, we treated mut _KRAS_ LUAD cell lines (_n_ = 6) with a pharmacological inhibitor to AURKB, Barasertib, which unlike PRKCH and PRKCA inhibitors reported to date, targets no additional kinases. We found a consistent synergistic effect in combination with the MEKi (Fig. 3K), suggesting that the combined Trametinib and Midostaurin effect may be in part mediated by AURKB inactivation.

**The KRASG12Ci Sotorasib replaces Trametinib with similar synergistic effect in vitro**

Next, we posited that a KRASG12Ci could replace Trametinib to cooperate with Midostaurin in mut _KRAS_ LUAD. We first performed cell

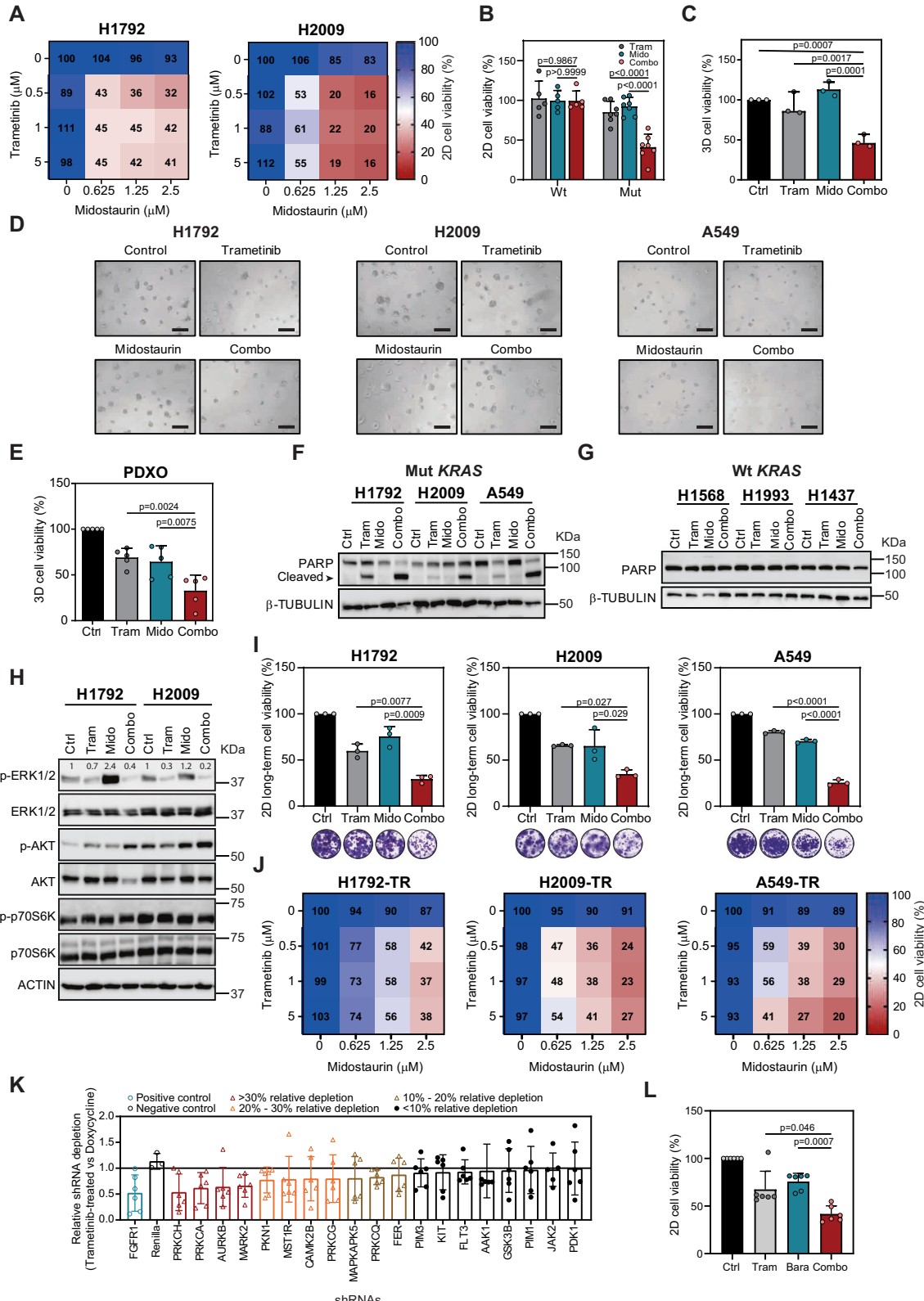

viability assays in *KRASG12C* cell lines with different sensitivity to the KRASG12Ci AMG-510 (Sotorasib) (Suppl. Fig. 4A). We found that the Sotorasib and Midostaurin combination had a more profound antiproliferative effect than single drugs on all human cell lines at 72 h and 5 days post-treatment (Fig. 4A and Suppl. Fig. 4B, C). The use of a second KRASG12Ci, Adagrasib, showed a similar effect in combination with Midostaurin (Suppl. Fig. 4D). The drug combination displayed no

activity in non-*KRASG12C* LUAD cell lines (Suppl. Fig. 4E). Interestingly, mouse LUAD cells derived from *KRas*[FSFG12C]; *P53*[FRT/FRT] mice were sensitive to the dual treatment at doses where single drugs had no or minor antiproliferative effect (Suppl. Fig. 4F). Furthermore, combined Sotorasib and Midostaurin treatment in human *KRASG12C* LUAD cells grown as 3D cultures or tumor organoids from LUAD PDXs had a larger antiproliferative effect compared to single drugs (Fig. 4B, C). Direct

**Fig. 3 | Effect of Trametinib and Midostaurin combination on *KRAS*-mutated LUAD cells. A** H1792 and H2009 percent cell viability after Trametinib (Tram) and Midostaurin (Mido) treatment for 72 h. **B** Effect of Tram and Mido combination on cell viability of wild-type (wt: H1437, H2126, H1568, H1993 and H1650) and mutant (mut: H1792, H2009, A549, HCC44, H23, H358 and CP435) *KRAS* LUAD cells (72-h treatment). Tram: 0.5 µM; Mido: 0.625 µM (data: mean +/− SD; test: one-way ANOVA, Tukey's adjustment). **C** Effect of Tram and Mido combination on cell viability of mut *KRAS* cells (H1792, H2009 and A549) in 3D (72-h treatment; *n*: 3 independent experiments). Tram: 0.01–0.05 µM; Mido: 0.05 µM (data: mean +/− SD; test: one-way ANOVA, Tukey's adjustment). **D** Representative images of H1792, H2009 and A549 3D (72-h treatment). Scale bar: 200 µm. **E** Effect of Tram and Mido on mut *KRAS* patient-derived xenograft organoids (PDXOs: TP60, TP69, TP80, TP181 and TP126; 72-h drug treatment). Tram: 0.05 µM; Mido: 0.3 µM (data: mean +/− SD; test: one-way ANOVA, Tukey's adjustment). **F, G** Cleaved PARP expression in mut (H1792, H2009 and A549; **F**) and wt *KRAS* (H1568, H1993 and H1437; **G**) cell lines after drug exposure (24-h treatment; loading control: β-TUBULIN). **H** Western blot of indicated proteins in H1792 and H2009 (48-h treatment; loading control: ACTIN). **I** Long-term assays of H1792, H2009, and A549 cells (10-day treatment; *n*: 3 independent experiments; data: mean +/− SD; test: one-way ANOVA, Tukey's adjustment). Crystal violet-stained images of control, Tram- (5 nM), Mido- (100 nM) and combo-treated cells. **J** Percent cell viability of Tram-resistant (TR) H1792, H2009 and A549 cells (72-h treatment). **K** Relative depletion of shRNAs against Midostaurin targets in Trametinib-treated versus doxycycline-treated H23 cells (n: 6 shRNAs/gene). Blue: hit selected by Manchado et al.[7]. Red: hits sensitizing to Trametinib with >30% relative depletion. Orange: hits sensitizing between 20% and 30%. Brown: hits sensitizing between 10% and 20%. **L** Effects of Tram (10–50 nM) and Barasertib (Bara; 25–1000 nM) combination on cell viability of mut *KRAS* (H1792, H2009, A549, HCC44, H23, H358) cells (72-h treatment; data: mean +/− SD; test: Kruskal–Wallis, Dunn's adjustment).

comparison of the Midostaurin-Sotorasib combination with KRASG12Ci-based combinations being tested in clinical trials (Sotorasib-Afatinib or Sotorasib-Trametinib) showed equal or more effective antiproliferative responses in 2D and 3D (Fig. 4D, E and Suppl. Fig. 4G, H).

Next, we assessed Sotorasib and Midostaurin treatment in the context of resistance. Using clonogenic assays, we found that the drug combination had a stronger impact on cell proliferation than single drug treatments, thus abrogating early adaptive resistance mechanisms (Fig. 4F, G). Second, using *KRASG12C* LUAD cell lines made resistant to the KRASG12Ci (Suppl. Fig. 4I), we observed that they still remained sensitive to the Sotorasib and Midostaurin combination (Fig. 4H). These results suggest that Sotorasib plus Midostaurin may represent an adequate therapeutic strategy for *KRASG12C* LUAD.

To assess the molecular mechanisms underlying the effect of the drug combination, we used intrinsically KRASi-resistant LUAD cell lines (H1792 and HCC44; Suppl. Fig. 4A). The dual treatment elicited a larger apoptotic response in cells treated with the combo condition with regard to single agents, suggestive of an enhanced cytotoxic effect (Fig. 4H). No consistent downregulation of canonical KRAS downstream kinases (p-ERK, p-AKT and p-p70S6K) or additional compensatory mechanisms that could explain the enhanced sensitivity to the dual treatment was seen (Fig. 4I). As expected, an electrophoretic mobility shift of KRASG12C protein band migration, indicative of covalent modification of mut *KRAS*, was detected by Western blot. Notably, addition of Midostaurin increased KRASG12Ci engagement in cells where target binding was not complete.

To discern the role of Midostaurin targets sensitizing to Trametinib in the context of KRASG12C inhibition, we carried out pharmacological experiments and found that combined KRASG12C and AURKB blockade has a stronger antiproliferative effect than single treatments (Fig. 4J). These observations suggest common mechanisms of action between MEK1/2 and KRASG12C inhibitors when combined with Midostaurin.

### Antitumor effect of Midostaurin-based drug combinations in treatment naive and resistant mut *KRAS* LUAD

First, 10 to 12-week old immunodeficient (Rag2[-/-]; Il2γr[-/-]) mice carrying cell-derived xenografts (CDXs) from parental H1792 and A549 cells were treated with Trametinib, Midostaurin or both when an average volume of ~100 mm[3] was reached. The drug combination induced the highest antitumor effect in the two models compared to single drug or vehicle treatment (Fig. 5A, C), and yielded more tumor regressions (13 out of 16) than single treatments (0 out of 16 for Trametinib and 1 out of 16 for Midostaurin) (Fig. 5B, D). No overt changes in mouse weight were observed (Suppl. Fig. 5A, B). We also tested the consequences of Trametinib and Midostaurin treatment in CDX models generated from TR cells. Tumors derived from H1792-TR cells were confirmed to be more resistant to the MEKi than those tumors arising from parental

cells (Suppl. Fig. 5C, D). Of note, combined administration of Trametinib and Midostaurin impaired tumor growth as compared to Trametinib alone administration albeit not as overtly as in Trametinib-naïve tumors (Fig. 5E, F). Mouse weight was also not affected by the treatments (Suppl. Fig. 5E).

Second, we investigated the impact of combined Sotorasib and Midostaurin treatment. Tumors from H358 and H1792 cells were generated in immunodeficient mice, which were administered single and double treatments. Midostaurin had no effect on tumor growth while Sotorasib elicited a short antitumor response that was lost upon continuous treatment. By contrast, dual Sotorasib and Midostaurin administration yielded tumor regressions that were sustained until treatment termination (Fig. 5G–J). Relevantly, the dual treatment also compromised tumor growth in tumors derived from Sotorasib-resistant cells (H358-SR) (Fig. 5K, L). Neither changes in mouse weight nor liver morphology, the latter determined by structural changes or anomalous infiltration of immune cells, were observed in single or double treatments (Suppl. Fig. 5G–I).

Based on the CDXs results, we decided to test the Midostaurin and Sotorasib combination in a lung cancer model driven by KRasG12C (*KRas*[FSFG12C]; *P53*[FRT/FRT] mice)[46]. We prioritized this combination over the one with Trametinib given that Sotorasib administration has been recently approved by the FDA for lung cancer treatment as single agent. In this GEM model, Sotorasib elicits either stable disease or partial response in approximately three quarters of tumors at 4 weeks of treatment[46], similar to the antitumor response pattern reported for lung cancer patients in the CodeBreaK100 clinical trial[47]. In keeping with this data, we observed tumor growth abrogation or regression at 4 weeks in 5 out of 8 tumors treated with Sotorasib, while no antitumor effect was found for Midostaurin. However, most Sotorasib treated tumors relapsed by week six, suggesting the development of resistance mechanisms (Fig. 6A–C). By contrast, the drug combination elicited antitumor responses that were sustained, or even increased, over time until the end of the treatment (Fig. 6A–C), an effect that was not attributable to differences in tumor volume at the beginning of the experiment (Suppl. Fig. 5J). Additionally, treatment with the drug combination reduced weight loss linked to disease progression (Suppl. Fig. 5K). MicroCT follow up revealed that new tumors became detectable on weeks two through six after the treatment was started exclusively in the vehicle- (*n* = 5), Midostaurin- (*n* = 1) and Sotorasib-treated (*n* = 3) groups. Furthermore, the follow up of some tumors by microCT was precluded by lung inflammation or collapse in the control, Midostaurin or Sotorasib groups but not in the drug combination one. Despite mice treatment for 6 weeks, no liver changes were detected (Suppl. Fig. 5L). These results indicate that a drug combination consisting of Midostaurin administration with either MEKi or KRASG12Ci may be efficacious for the treatment of mut *KRAS* LUAD, particularly in treatment-naïve tumors.

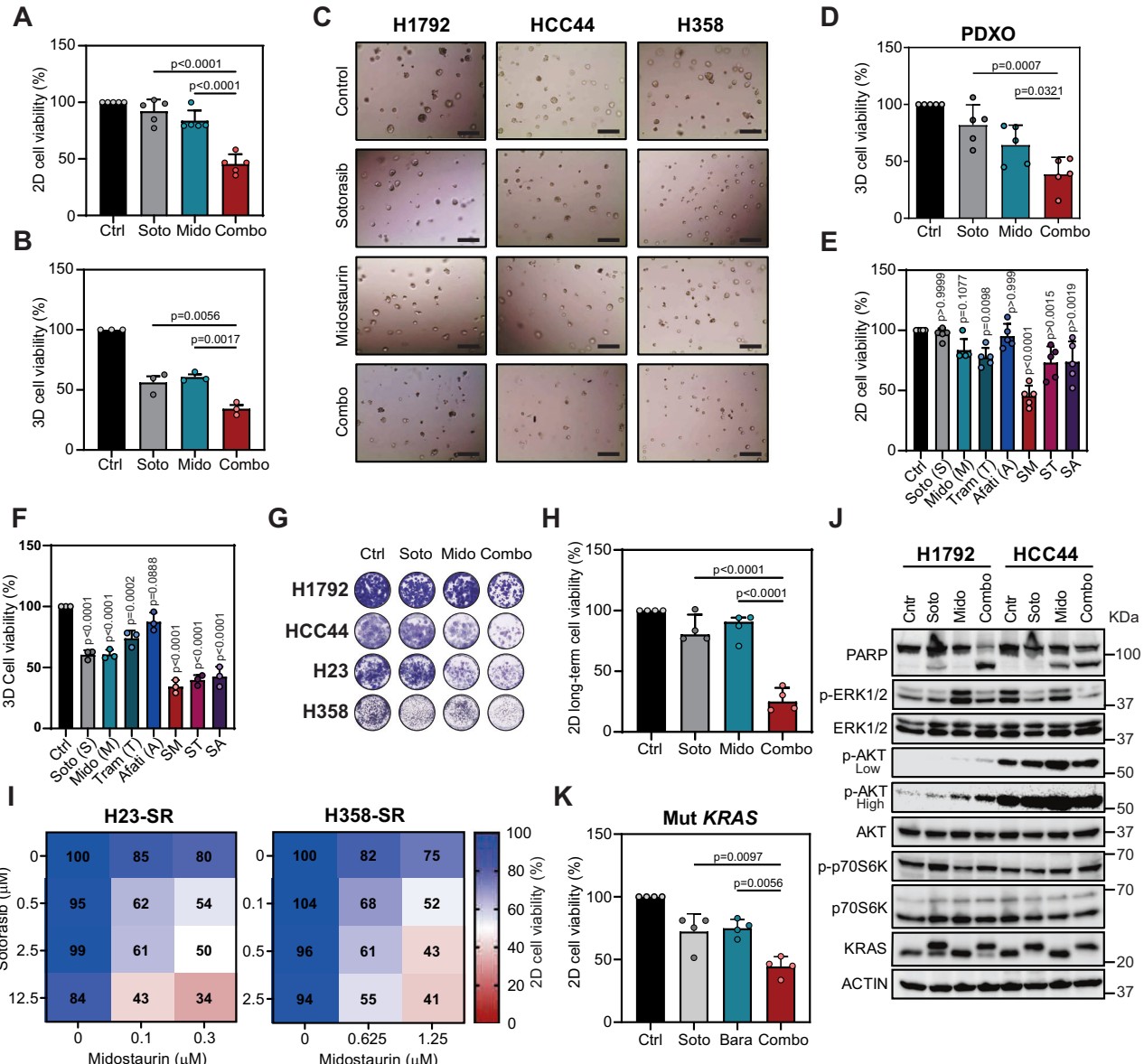

**Fig. 4 | Synergistic effect of KRASi Sotorasib and Midostaurin combination on *KRASG12C* LUAD cells. A** Cell viability of *KRASG12C* LUAD cells (H1792, HCC44, H23, H358 and CP435) treated with Sotorasib (Soto; 20–500 nM), Midostaurin (Mido; 0.3–1.25 μM) or both (72-h treatment; data: mean +/− SD; test: one-way ANOVA, Tukey's adjustment). **B** Cell viability of *KRASG12C* cells (H1792, HCC44 and H358) in 3D (72-h treatment). Soto: 62.5 nM; Mido: 0.3 μM (data: mean +/− SD; test: one-way ANOVA, Tukey's adjustment). **C** Representative images of H1792, HCC44 and H358 3D cultures exposed to treatments. Scale bar: 200 μm. **D** Effects of Soto and Mido combination on cell viability of mut *KRAS* patient-derived xenograft organoids (PDXOs: TP60, TP69, TP80, TP181, TP126) in 3D (72-h treatment). Soto: 62 nM; Mido: 0.3 μM (data: mean +/− SD; test: one-way ANOVA, Dunnet's adjustment). **E** Percent cell viability of *KRASG12C* cells (H1792, HCC44, H23, H358 and CP435) treated with Soto (S), Mido (M), Tram (T), Afatinib (Afati, A) or combinations for 72 h (data: mean +/− SD; test: oneway ANOVA, Bonferroni's adjustment).

**F** Percent cell viability of *KRASG12C* cells (H1792, HCC44, and H358) grown in 3D, after 72 h treatment with Soto (S), Mido (M), Tram (T), Afati (A) or indicated combinations (data: mean +/− SD; test: one-way ANOVA, Bonferroni's adjustment). **G** Representative crystal violet staining images of 10-day treatment (control: Ctrl; Soto: 12,5-62,5 nM; Mido: 50–100 nM). **H** Long-term effect of Soto and Mido combination on cell viability of *KRASG12C* cells (H1792, HCC44, H23 and H358). 10-day treatment (data: mean +/− SD; test: one-way ANOVA, Tukey's adjustment). **I** Cell viability percentage of Soto-resistant (SR) H23 and H358 cells treated with indicated concentrations of Soto and Mido, individually or in combination (average of 3 experiments). **J** Western blot analysis of indicated proteins in H1792 and HCC44 *KRASG12C* cells (48-h treatments; loading control: ACTIN; exposure time: low and high). **K** Effects of Soto and Barasertib (Bara) combination on cell viability of mut *KRAS* (H1792, HCC44, H23, H358) cells (72-h treatment; Soto: 0.02 − 5 μM; Bara: 5 μM; (data: mean +/− SD; test: one-way ANOVA, Tukey's adjustment).

To test the potential implication of the tumor microenvironment in response to combined Midostaurin and Sotorasib treatment, we developed a flexible syngeneic model using T1 cells derived from the *KRas*FSFG12C; *P53*FRT/FRT lung cancer model via subcutaneous injection in F1 C57BL/6 x 129S4/Sv mice. The antitumor effect of the drug combination was consistent with the GEM model even after only 7 days of treatment (Suppl. Fig. 5M). At this time point, we analysed CD8 + T lymphocytes and observed an overt infiltration into the tumor site (Fig. 5D, E, and Suppl. Fig. 5N). These findings suggest that non-cell autonomous effects involving immune cell populations may also contribute to the antitumor effect of the drug combination.

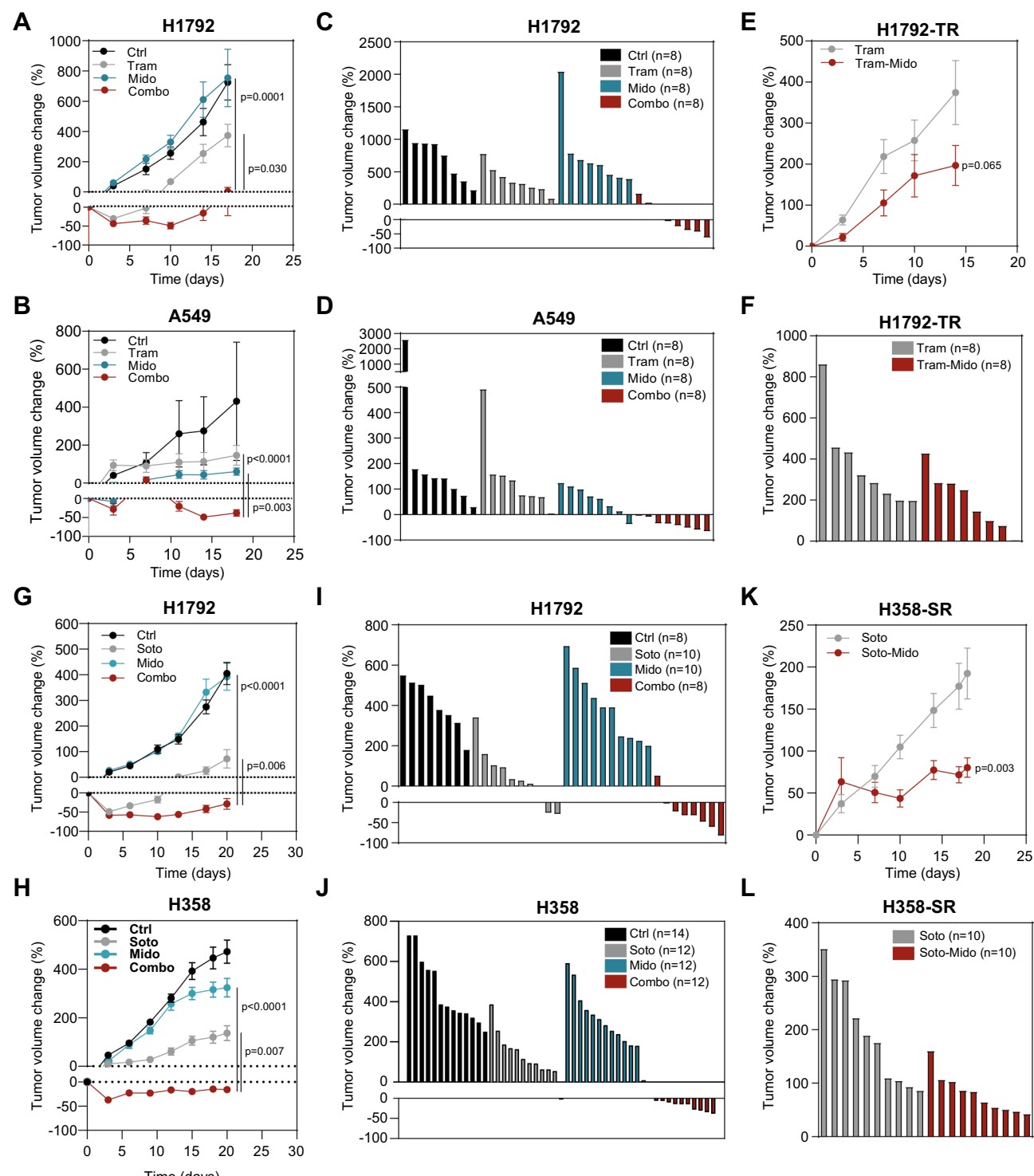

**Fig. 5 | Midostaurin-based drug combinations show antitumor effects on treatment naïve and resistant mut *KRAS* LUAD. A**, **B** Percent fold change growth of cell-derived tumors (CDXs) from H1792 or A549 cells treated with indicated drugs (Trametinib: 1 mg/kg; Midostaurin: 25 mg/kg). *N* = 8 tumors per group in Rag2[-/-]; Il2γr[-/-] mice (data: mean +/− SEM; test: Kruskal-Wallis, Dunn's adjustment). **C**, **D** Waterfall plots of tumors from (**A**) and (**B**) at the last day of experiment. **E** Percent fold change of Tram-resistant (TR) H1792 CDXs treated with indicated drugs (Trametinib: 1 mg/kg; Midostaurin: 25 mg/kg). *N* = 8 tumors per group in Rag2[-/-]; Il2γr[-/-] mice (data: mean +/− SEM; test: Mann−Whitney). **F** Waterfall plot of

tumors from (**E**) at the last day of experiment. **G**, **H** Percent fold change of CDXs from H1792 (**G**) or H358 (**H**) treated with indicated drugs. 30 mg/kg (H1792) and 10 mg/kg (H358); Mido: 25 mg/kg. *N*: 14 (ctrl) and 12 (Soto, Mido and Combo) tumors in Rag2[-/-]; Il2γr[-/-] mice data: mean +/− SEM; test: one-way ANOVA, Tukey's adjustment). **I**, **J** Waterfall plot of tumors in (**G**) and (**H**) at the last day of experiment. **K** Percent fold change growth of Soto-resistant (SR) H358 CDXs treated with indicated drugs (Soto: 10 mg/kg; Mido: 25 mg/kg). *N*: 10 tumors per group in Rag2[-/-]; Il2γr[-/-] mice (data: mean +/− SEM; test: *t*-test). **L** Waterfall plot of tumors from (**K**) at the last day of experiment.

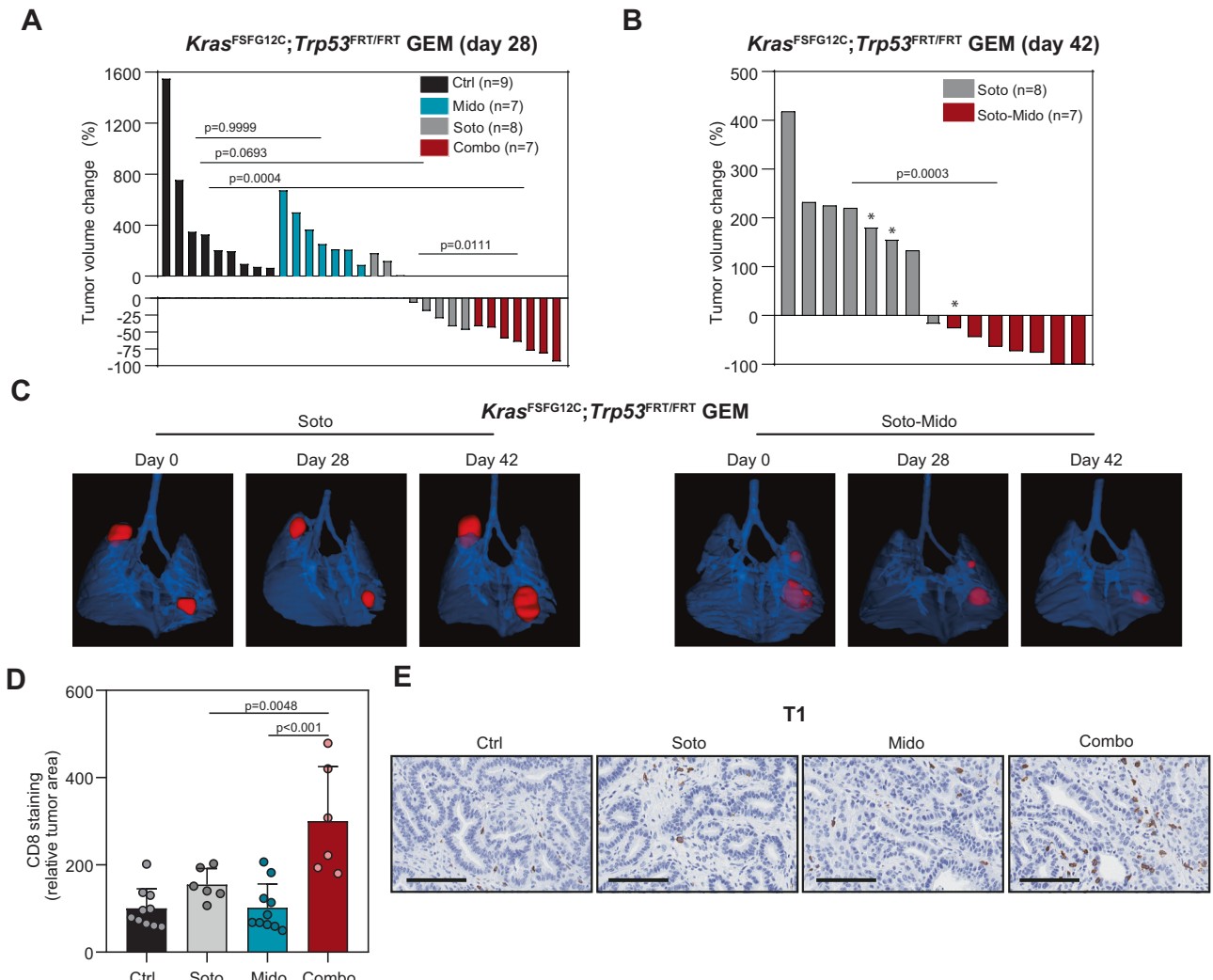

**Fig. 6 | Midostaurin potentiates the antitumor effect of Sotorasib in a GEM lung cancer model. A** Waterfall plot of lung tumors from the *KrasG12C* driven model (*Kras*^FSFG12C^; *Trp53*^FRT/FRT^ mice) treated with the indicated drugs at day 28. *N*: number of tumors per group. Soto: 100 mg/kg; Mido: 25 mg/kg (test: Kruskal–Wallis, Dunn's adjustment -all comparisons-; *t*-test -Soto vs combo-). **B** Waterfall plot graph of lung tumors from *Kras*^FSFG12C^; *Trp53*^FRT/FRT^ mice at day 42 of treatment. *: units correspond to tumor volume change at day 28. *N*: number of tumors per group. Soto: 100 mg/

kg; Mido: 25 mg/kg (test: Mann–Whitney). **C** Representative microCT images of lung tumors from *Kras*^FSFG12C^; *Trp53*^FRT/FRT^ mice at different days of treatment. **D** Quantification of CD8 staining in tumors from T1-derived xenografts in F1 C57BL/6 x 129S4/Sv mice (7-day treatment). *N*: 10 tumor sections (ctrl and Mido); n: 6 tumor sections (Soto and Combo). Soto: 30 mg/Kg; Mido: 25 mg/kg (data: mean +/− SEM; test: one-way ANOVA, Tukey's adjustment). **E** Immunohistochemistry images of tumors in (**D**) stained for CD8. Scale bar: 100 μM.

## MtPKCi-based drug combinations result in MYC protein decrease

To gain a global mechanistic view on the consequences of dual treatments, tandem-mass-tags (TMT) was applied to quantitate the proteome of H1792 cells treated with Trametinib, Lestaurtinib and both drugs. Whole proteome studies unveiled 279, 102, and 98 dysregulated proteins in Trametinib, Lestaurtinib and dual treatment conditions relative to untreated cells (± 20% change expression, *p* < 0.05) (Suppl. Data 2). Most of the protein changes induced by the three treatments were specific to each condition (range 65–80%) (Suppl. Fig. 7A). In particular, 80 out of 98 proteins were significantly dysregulated in the combination condition with regard to single drugs (*p* < 0.05). Interrogation of the biological pathways associated with dysregulated proteins elicited by the drug combination unveiled several enriched pathways including ribosomal scanning and start codon recognition, RHO GTPase effector, regulation of actin cytoskeleton, positive regulation of binding, MYC active pathway and regulation of protein polymerization (Log*P* < −5) (Fig. 7A). Complementary enrichment analysis of the downregulated protein set using Molecular Signature

Data Base (MSigDB) unveiled MYC targets as the most enriched gene set (Suppl. Fig. 6B, C). Indeed, several genes downregulated upon the dual treatment, such as *XPO1* and *GLS*, previously reported as oncogene vulnerabilities in mut *KRAS* lung cancers[48,49], are MYC targets. PPI analysis revealed that XPO1 was linked to both nucleoporin complex components (NUP155 and NUP188) and the nuclear importin KPNB1 (Suppl. Fig. 6D), suggesting the involvement of the nuclear import-export system in the response to the dual treatment.

The previous data led us to analyse MYC expression. Trametinib and Lestaurtinib combination decreased MYC expression to a greater extent than single drugs in both H1792 and H2009 (Fig. 7B). Similar results were found in mut *KRAS* H1792 and HCC44 cells treated with the Trametinib and Midostaurin combination (Fig. 7C). We noted no changes in MYC mRNA expression levels across single or dual treatments at 24 and 48 h that could account for the MYC protein changes observed (Suppl. Fig. 6E, F). MYC stability can be regulated post-transcriptionally through proteasome degradation. However, proteasome inhibition in two independent cells exposed to Trametinib plus Midostaurin did not rescue MYC levels (Suppl.

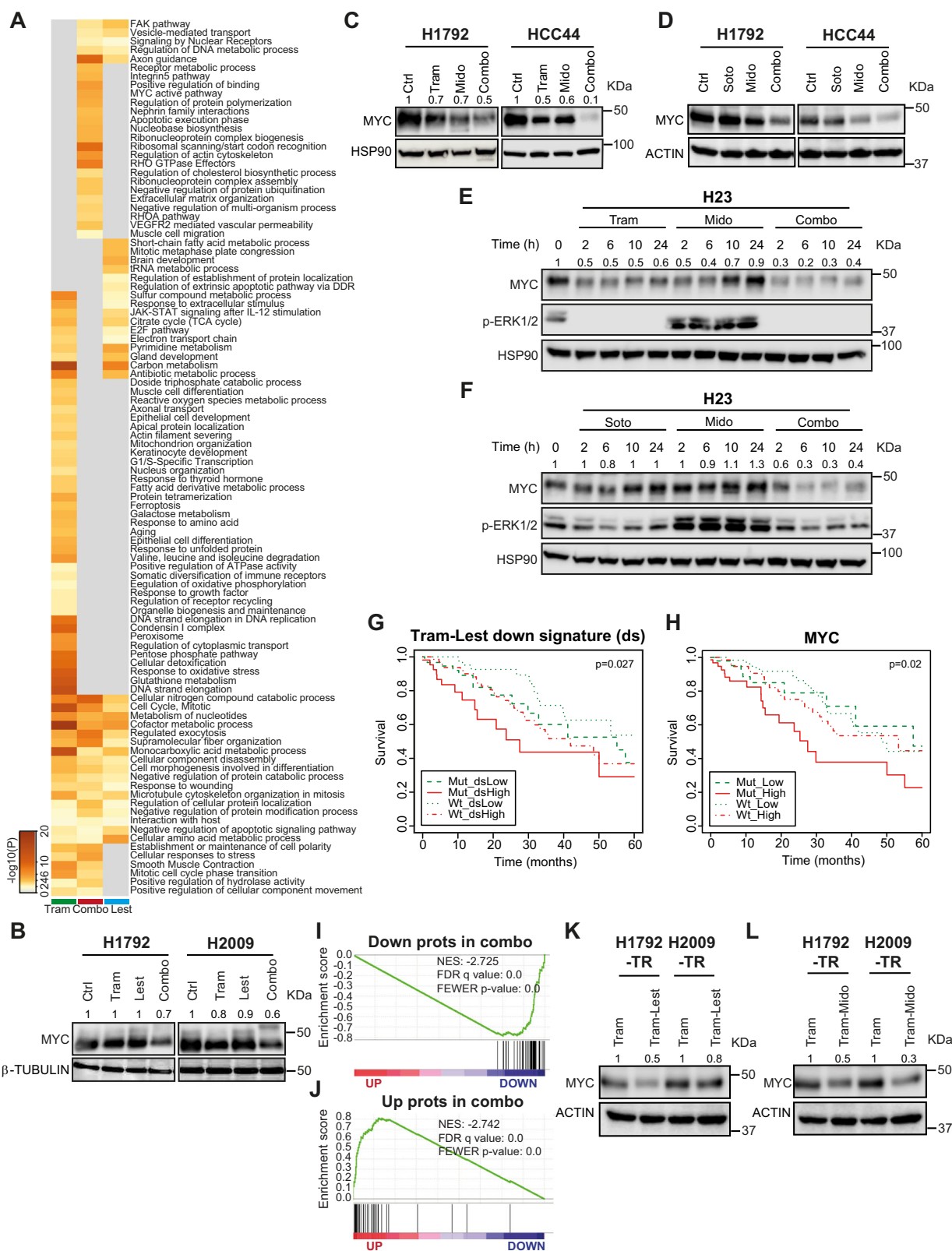

Fig. 6G), suggesting alternative post-transcriptional mechanisms to MYC downregulation. Given the coinciding antitumor effect involving combination of mtPKCi with MEK1/2 or KRASG12C inhibitors, MYC expression was assessed in the context of the KRASG12Ci-based drug combination. MYC was particularly downregulated in those cells exposed to concomitant Sotorasib and Midostaurin administration (Fig. 7D). Time course analysis of MYC expression

kinetics revealed impaired MYC expression in response to combined Trametinib-Midostaurin or Sotorasib-Midostaurin treatments compared to single treatments as early as 2 h after treatment (Fig. 7E, F and Suppl. Fig. 6H, I). MYC decrease was also observed in tumors derived from T1 cells exposed to the drug combination (Suppl. Fig. 6J, K). Thus, MYC may be involved in the response to both Midostaurin-based combinations.

**Fig. 7 | MtPKCi-based drug combinations downregulate MYC protein. A** Heatmap of biological pathways enriched by the dysregulated proteins obtained from H1792 cell line 48 h after exposure to Trametinib (Tram), Lestaurtinib (Lest) or both, and generated by METASCAPE[70]. **B, C** MYC protein expression of H1792 and H2009 (**B**) or H1792 and HCC44 (**C**) cell lines treated for 48 h. β-TUBULIN (**B**) or HSP90 (**C**) are loading controls. Numbers correspond to relative MYC densitometry quantification. **D** MYC protein expression of H1792 and HCC44 cell lines treated for 48 h (loading control: ACTIN). **E, F** MYC and p-ERK1/2 protein expression of H23 cell line at different time points (loading control: HSP90). Numbers correspond to relative MYC densitometry quantification. **G, H** Kaplan–Meier plot showing overall survival of lung cancer patients from TCGA database as a function of the expression of the genes from the signature downregulated by the Tram and Lest combination (ds) and KRAS mutational status (**G**) or MYC expression and KRAS mutational status (**H**). **I** Gene set enrichment analysis (GSEA) of proteins downregulated by combined Tram and Lest treatment onto a ranked-order list from H1792-TR exposed to Tram compared to those cells treated with the drug combination. **J** GSEA of proteins upregulated by combined Tram and Lest treatment onto the same ranked-order list from (**I**). **K, L** MYC protein expression in Tram-resistant H1792 and H2009 cell lines after 48 h drug treatment (loading control: ACTIN). Numbers correspond to relative MYC densitometry quantification.

To test the clinical relevance of the molecular findings, we queried the association of the signature downregulated by the drug combination as well as of MYC itself with patient survival. High expression levels of either the downregulated signature or MYC along with *KRAS* mutations identified patients with the worst survival outcome (Fig. 7G, H), providing indirect evidence to support a functional role in patients.

We next investigated the molecular mechanisms underlying the response of Trametinib-resistant (TR) mut *KRAS* LUAD cells to the drug combination, as they responded similarly to parental cells. H1792-TR treated with Trametinib or the drug combination were submitted to proteome profiling. Thirty-eight up and 42 downregulated proteins were obtained (Suppl. Data 2). GSEA analysis was performed by querying the downregulated and upregulated protein sets elicited by the dual treatment in parental cells against the ranked-ordered list derived from the proteome analysis in TR cells treated with Trametinib and Lestaurtinib. A strong negative and positive enrichment was found for each protein set respectively (Fig. 7I, J). This finding prompted us to investigate MYC protein levels in TR cells exposed to the drug combination. A decrease in MYC expression was found compared to Trametinib-treated ones (Fig. 7K). Similar observations were made when Midostaurin was used in the drug combination (Fig. 7L). These data suggest that parental and TR mut *KRAS* LUAD cells share overlapping mechanisms of response to combined Trametinib and mtPKCi treatment.

## MYC upregulation increases resistance to MEK and KRASG12C inhibitors

The previous results suggest that MYC may be a mediator of dual treatments' antitumor effect. One potential explanation is that MYC expression may represent a resistance mechanism to single treatments. To address this question, we took advantage of TR cells. Proteomics analysis of parental and TR cells revealed upregulation and downregulation of 90 and 98 proteins respectively (Suppl. Data 2). We first inquired the clinical role of TR dysregulated proteins in human cancer using survival analysis. We focused on upregulated proteins to Trametinib treatment, as they could have a pro-oncogenic role. Patients with high expression of the upregulated signature had the worst survival outcome when *KRAS* was mutated (Fig. 8A), suggesting a likely role in the disease. Second, analysis of the biological pathways enriched in the dysregulated protein signature revealed a MYC active pathway in resistant cells (Fig. 8B). In accordance, MSigDB analysis unveiled MYC as a potential transcriptional regulator of the increased gene signature (Suppl. Fig. 7A).

These results led us to characterize MYC expression in TR cell lines. MYC upregulation was seen compared to parental cells and occurred even in the presence of MEK1/2 inhibition (Fig. 8C), suggesting ERK1/2 independent MYC regulatory mechanisms. MYC overexpression was related to enhanced basal mRNA transcription (Fig. 8D). In support of this observation, MYC phosphorylation levels were nearly identical to MYC endogenous levels (Fig. 8D) and, thus, unlikely to contribute to further MYC increase in the absence of ERK1/2 activation.

To directly test the functional implication of MYC in MEKi resistance, exogenous MYC was expressed in parental mut *KRAS* cells with various endogenous MYC levels (H2009 and H358) (Fig. 8E and Suppl. Fig. 7B). MYC overexpression rendered parental cells more resistant to Trametinib (Fig. 8F, G), indicating a functional link to the resistance phenotype. Lastly, given the similar involvement of MYC in the response to combined Sotorasib and Midostaurin and, more importantly, that MYC amplification has been reported in lung cancer tumors acquiring resistance to Sotorasib[50], we tested MYC function in the context of KRASG12Ci treatment. Paralleling the Trametinib results, exogenous MYC overexpression enhanced resistance to Sotorasib (Fig. 8H). Notably, this result was mimicked in vivo (Fig. 8I). These results suggest that MYC upregulation contributes to the resistant phenotype to both MEK1/2 and KRASG12C inhibitors.

## Discussion

Combined inhibition of KRAS or its proximal effectors along with feedback pathways is a potential efficacious strategy to treat *KRAS*-mutated lung cancer. Selection of targets for the optimization of therapeutic strategies has been mainly inspired by their direct involvement in KRAS oncogenesis or resistance to targeted therapies. Here, we have capitalized on the distal output of *KRAS* oncogene to initially unveil the combination of MEK1/2i and mtPKCi as a potential dual strategy for LUAD treatment. Our results provide the proof-of-principle on the integration of gene signature-based drug repurposing approaches and pairwise drug screening as a feasible strategy to identify novel combinatorial strategies in cancer. Therefore, this approach to nominate drugs entering combinatorial strategies provides a rational framework for the discovery of new therapies[1], what may become especially relevant for other cancers lacking efficacious treatments.

Resistance mechanisms to MEK1/2 inhibition in mut *KRAS* lung cancer involve activation of distinct signaling elements within the KRAS network, such as PI3K, BCL-XL, STAT3, IGFR, EGFR, FGFR1, JNK, ERK or SHP2[4–12]. Beyond SHP2 activation, which seems to be the most general adaptive mechanism, the participation of other pathways circumscribes to subsets of lung cancers. Thus, finding therapeutic strategies for a large fraction of lung cancer patients is of paramount importance. Our results, based on human and mouse cell lines, suggest that the MEKi and mtPKCi combination may be effective in a large spectrum of mut *KRAS* lung cancers regardless of the type of *KRAS* mutation or additional concurrent alterations in tumor suppressor genes. However, it is plausible that other *KRAS* mutations beyond those studied in this work may have dissimilar sensitivity to the treatment. Experiments in a larger panel of cell lines spanning all *KRAS* mutations found in lung cancer will aid to resolve the overall involvement of KRAS in treatment response.

KRASG12C inhibitors have reached clinical trials with enormous expectations and the results obtained in lung cancer led to the recent approval of Sotorasib by the FDA[13,14]. Nonetheless, both the clinical findings and experimental data illustrate the need for combinatorial strategies to maximize antitumor efficacy[13,14,18,19]. So far, combinatorial approaches based on KRASi incorporate drugs already tested in combination with MEKi, under the premise that KRASi will yield better antitumor responses and decreased toxicity. However, recent data suggests that adaptation mechanisms to KRAS inhibition are cell-type

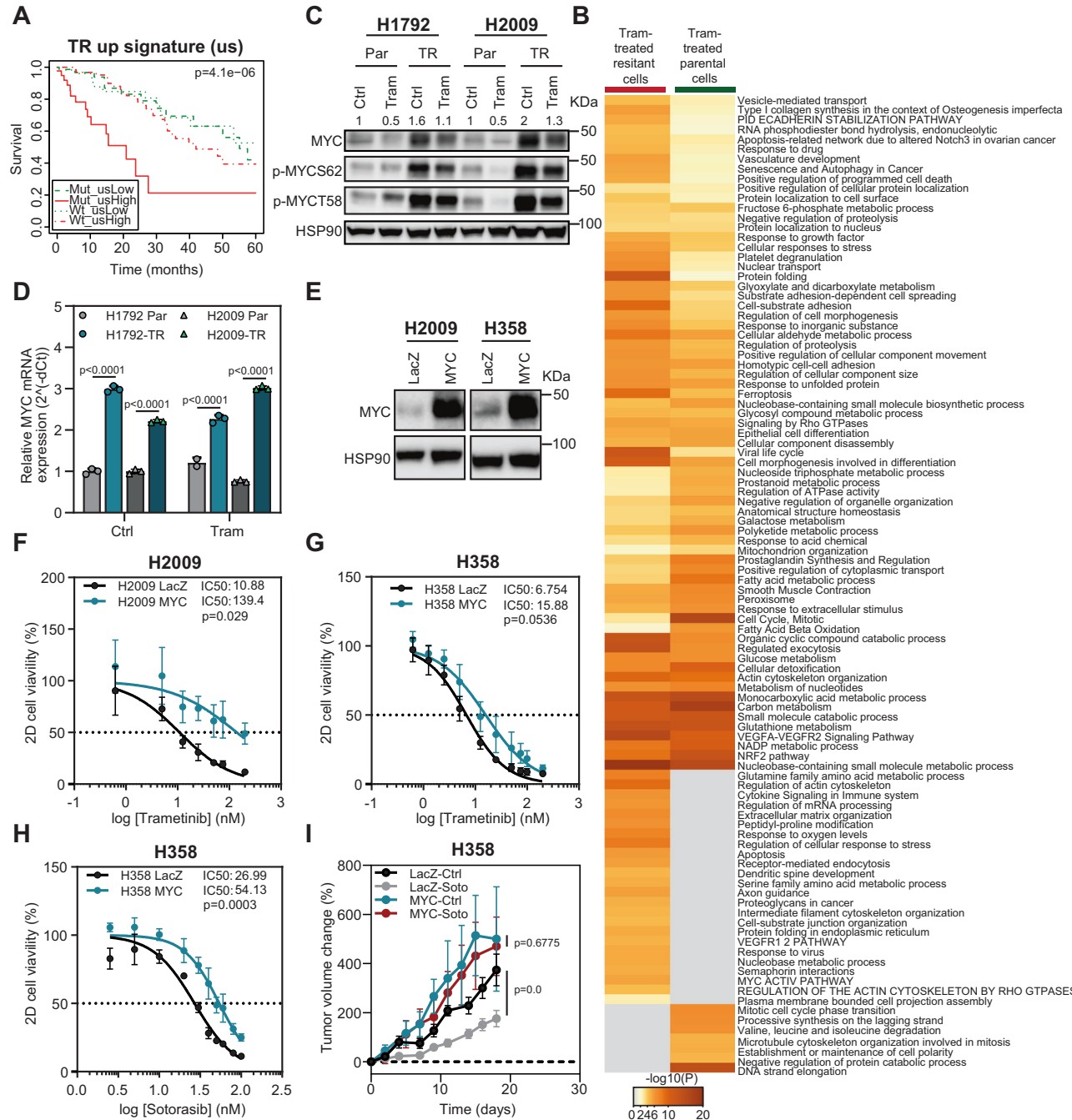

**Fig. 8 | MYC upregulation contributes to MEKi and KRASi resistance.**
**A** Kaplan–Meier plot showing overall survival of lung cancer patients from TCGA database as a function of the expression of genes from the Trametinib-resistant upregulated signature (us) and KRAS mutational status. **B** Heatmap of biological pathways enriched by the dysregulated proteins obtained from Tram-treated parental and Tram-resistant (TR) H1792 cells 48 h after exposure to Trametinib (Tram) using METASCAPE[70]. **C** MYC, p-MYC S62 and p-MYC T58 proteins' expression of Parental (Par) and TR H1792 and H2009 cell lines 48 h after exposure to indicated treatments. Ctrl: Control; Tram: Trametinib (loading control: HSP90). Numbers correspond to relative MYC densitometry quantification. **D** Relative MYC mRNA expression of Parental (Par) and Trametinib-resistant (TR) H1792 and H2009 cell lines 24 h after Trametinib treatment. Housekeeping gene for qPCR: GAPDH (*n*: 3 independent experiments; data: mean +/− SD; test: two-way ANOVA, Sidak's

adjustment). **E** MYC protein expression in LacZ or MYC overexpressed H2009 and H358 cell lines (loading control: HSP90). **F, G.** Percent cell viability of LacZ or MYC overexpressed H2009 and H358 cells after 10-day treatment with increasing concentrations of Trametinib. The IC50 values for Trametinib are shown for each cell line. *P* values were obtained from the comparison of IC50 values between control (LacZ) and MYC-overexpressing cells (*n* = 4 experiments; data: mean +/− SD; test: Mann–Whitney -H2009- and *t*-test -H358-). **H** Percent cell viability of LacZ- or MYC-overexpressing H358 cells after 10-day treatment with increasing concentrations of Sotorasib. IC50 values for Sotorasib are shown for each cell line. *P* values were obtained from the comparison of IC50 values between control (LacZ) and MYC-overexpressing cells (*n* = 4 experiments; test: *t*-test). **I** Percent fold change growth of cell-derived tumors (*n* = 12) from LacZ- and MYC-overexpressing H358 cells treated with Sotorasib (Soto: 10 mg/Kg). (data: mean +/− SEM; test: *t*-test).

(epithelial vs. mesenchymal) dependent and that even similar cell types engage different compensatory mechanisms[20], highlighting the need to find drug combinations that span a large fraction of *KRASG12C* lung cancers. Our findings provide evidence that KRASi can replace MEKi in combination with mtPKCi not only with similar efficacy to previously described combinations but also across multiple cell lines reported to have distinct compensatory mechanisms to KRASG12C inhibitors.

While our study proposes the use of mtPKCi as KRASG12Ci partners, recently Santana-Codina et al. used proteomics signatures from mut *KRAS* cell lines to predict drugs that could deepen the effect of the KRASi ARS-1620 and unveiled Midostaurin (annotated as a FLT3i in that study) as a potential combinatorial partner[51]. Although the authors went on to test combinations with other predicted drugs such as mTOR or PI3K inhibitors, their study independently supports the selection of Midostaurin for KRASi-based combinatorial strategies. Combinations using Midostaurin have been previously explored, such as that with radiation and a standard chemotherapy in advanced rectal cancer (NCT01282502). With the recent approval of Sotorasib by the FDA, the development of a clinical trial in combination with Midostaurin could be foreseen, particularly since the reduced off-target effects of KRASG12Ci translate in a favourable safety profile that provides room for dual treatments. Nonetheless, the potential success of such dual strategy in the clinic will largely rely on the ability to limit additional toxicity in addition to blocking KRAS signaling output.

A better knowledge of the targets involved in response to multi-tyrosine kinase inhibitors is key to understand their mechanism of action. Our data strongly suggest that AURKB exerts a prominent role in the response to Trametinib and Sotorasib. Interestingly, AURKB inhibition enhances the effect of anti-EGFR therapies in LUAD[52,53]. Most importantly, and supporting their functional role in LUAD, high AURKB expression has been reported compared to normal tissue[54]. Nonetheless, we cannot exclude a contribution from kinases whose single blockade yields modest sensitization to Trametinib and Sotorasib but collectively render cells sensitive to MEK or KRASG12C inhibitors.

In connection with the potential participation of FLT3 in the response to mtPKCi-based combinations, FLT3 ligands foster mobilization of conventional dendritic cells (cDCs) in pancreatic ductal adenocarcinoma (PDAC) models to enhance CD8 + T cell and TH1 activity and, eventually, to reduce tumor growth and increase response to therapy[55]. While FLT3 inhibitors could then counteract this antitumor response, divergent T cell responses in LUAD with regard to PDAC caused by differences in cDC infiltration may argue otherwise. In addition, cDCs tend to be excluded from the tumor core to the tumor periphery in mouse PDAC but not in LUAD. Furthermore, increasing mobilization of cDC progenitors allows for favorable T cell responses in preneoplastic lesions but not in established PDAC models. Overall, these three arguments suggest that FLT3i should not impair cDCs function/mobilization in established LUAD mouse models. Indeed, the strong infiltration of CD8 + T cells to the tumor site observed in response to combined Sotorasib-Midostaurin treatment experimentally supports these ideas. Nonetheless, we cannot completely rule out that this effect may be different in patients.

At the molecular level, we noted the downregulation of the transcriptional regulator MYC in response to combined MEK1/2i and mtPKCi treatment, both in treatment-naïve and MEK1/2i-resistant cells. This mechanistic finding was also seen in *KRASG12C* LUAD cell lines when a KRASi was used in combination with a mtPKCi, highlighting additional common mechanisms involved in the response to MEK1/2 and KRASG12C inhibition. Downregulation of a MYC signature has been reported in vivo upon treatment of *KRASG12C* lung tumors with KRASi MRTX-849 using drug concentrations eliciting significant tumor growth inhibition[14]. Moreover, genetic MYC inhibition enhances MRTX-849 effect both in vitro and in vivo[14]. These results suggest that decreasing MYC activity may be required to yield antitumor drug

responses. Along these lines, MYC downregulation is needed for the effect of ERK1/2i in other mut *KRAS* tumors such as pancreatic cancer[56], supporting a role for MYC in maintaining tumor integrity in the context of targeted therapies to the RAS pathway. Thus, our results and those of others point at MYC downregulation as a means to achieve an optimal antitumor response in KRAS-driven cancers. Nonetheless, the participation of additional mediators in the mtPKCi-based combinations' response cannot be ruled out and will require further attention.

Complementarily, a potential role of MYC expression in the context of RAS pathway targeted therapies has been suggested by recent clinical data reporting MYC amplification in lung cancer tumors resistant to Sotorasib[50]. Our functional studies overexpressing MYC in *KRASG12C* lung cancer cells support an active role as a resistance mechanism to KRASi. In addition, while little clinical information is available on the resistance mechanisms to MEK1/2 inhibitors, our data suggests that high MYC levels may also contribute to the resistant phenotype, what occurs even in the absence of ERK1/2 reactivation. Taken together, these data suggest that MYC upregulation stands as a common resistance mechanism to different targeted therapies to the KRAS pathway. In light of these findings, one could expect that concomitant MYC and KRASG12C inhibition would yield strong antitumor responses, a hypothesis that could be now clinically tested by the use of the first-in-class MYC inhibitor, OMO-103[57], which has already been tested in an escalation dose phase I clinical trial with promising results (NCT04808362).

*KRAS* mutations are present in a wide fraction of tumors beyond those of the lung, including gastrointestinal tumors with dismal prognosis such as those of the pancreas and the colon. These mut *KRAS* cancers are often refractory to inhibition of the KRAS-BRAF-MEK pathway and preclinical data suggest that combinatorial pharmacological strategies involving KRAS pathway inhibitors may yield better antitumor responses[58–60]. This holds true even with the newly developed KRAS inhibitors, which require inhibition of alternative targets such as EGFR for deeper antitumor responses in colon cancer[61,62]. Given that the iKRASsig signature used for our drug repurposing approach was upregulated in both pancreatic and colorectal cancers[25], investigating the effect of MEKi- and KRASi-based combinatorial approaches involving mtPKCi's across KRAS-driven tumors may unveil unanticipated interventional opportunities for this type of tumors and, thus, expand our current armamentarium for these deadly cancers.

## Methods

### Ethics authorization
Our research complies with all relevant ethics regulations and the project was approved by the Research Ethics Committee (CEI) of the University of Navarra under the protocol number 2020.010.

### Drug repurposing study
To generate the Connectivity Scores between the interspecies signature and each chemical perturbagen (CP) available in the L1000 dataset[22], we used the Library of Integrated Network-based Cellular Signatures (LINCS) program (http://c3.lincscloud.org). For all analyses, we used the sig_query tool. To obtain predictions for every CP in the L1000 dataset, we set the flag --column_space to "full". To ensure maximum coverage of our signature, which due to high stringency was relatively small, we set the flag --row_space to "full". For all other settings we used defaults provided. Results were provided on the signature level, and thus showed the effect of a drug in one cell, at one dose, measured at one time-point. As there are multiple signatures per CP and we sought an overall assessment of CP effect in a novel condition, we first took the mean of scores within each cell line and then took the mean of those cell line means. This sequential averaging was done to avoid giving extra weight to effects in cells that had been more frequently assayed. Finally, to filter down to CPs with the most robustly assayed effects, we retained only CPs which had been measured in

more than 10 different conditions. All analysis and processing of results was done using R statistical programming language (https://www.R-project.org).

## Generation of an interspecies KRAS signature

The interspecies KRAS signature (iKRASsig) was obtained by integrating gene expression data from mouse and human experimental systems to uncover a core of genes consistently regulated by KRAS. Samples from three studies (GSE15325, GSE17671 and GSE49200) were normalized with robust multi-array average (RMA), the quality was evaluated and outlier detection with R/Bioconductor[63] was carried out. LIMMA (Linear Models for Microarray Data) was used to identify the probe sets with significant differential expression between experimental conditions. The KRAS gene set was defined as the genes with B > 0 and logFC > 1.5 or <0.75 in at least two of the three studied experimental models.

Gene set enrichment and survival analysis Enrichment of the obtained iKRASsig in human patients with mutated KRAS analyzed with Gene Set Enrichment Analysis (GSEA)[64]. Public datasets used for this analysis were downloaded from Gene Expression Omnibus (GEO) data repository (http://www.ncbi.nlm.nih.gov/geo) or The Cancer Genome Atlas (TCGA) Data Portal (https://tcga-data.nci.nih.gov/tcga/tcgaHome2.jsp). TCGA processed data for RNA-Seq experiments of LUAD samples and microarray raw data were downloaded for the logFC calculation of the comparison KRAS_mut vs KRAS_wt. Microarray raw data was also downloaded from GEO, normalized with RMA[65] and analyzed for the logFC calculation using R[66]. Survival analysis was conducted on both gene sets and individual genes using TCGA LUAD data (https://tcga-data.nci.nih.gov/tcga/tcgaHome2.jsp). Log-rank test was used to calculate the statistical significance of differences observed among Kaplan-Meier curves[67].

## Cell lines

Human wt (H1437, H1568, H2126, HCC78, H1650 and H1993) and mut *KRAS* (H1792, H2009, A549, H358, H23, HCC44, CP435) LUAD cell lines were used. All these cell lines but CP435, which was generated from a *KRASG12C* primary LUAD, were obtained from ATCC and authenticated by the Genomics Unit at CIMA using Short Tandem Repeat profiling (AmpFLSTR Identifiler Plus PCR Amplification Kit) in June 2016. Each cell line was expanded and biobanked. Subsequently, all cell lines used were obtained from the biobank and passaged for a maximum of 10 passages. Human cells were grown according to ATCC specifications. Mouse mut *KRas* cell lines derived from lung adenocarcinoma (LUAD) models (*KRasG12D*: KLA and KLAp53ko; *KRasG12C; Trp53ko*: T1, T2, T3; *KRasG12V; Trp53ko*: 220-1, 220-2, 95) were grown in DMEM supplemented with 10% serum and 1% penicillin/streptomycin. Human and mouse cell lines were tested for mycoplasma using the MycoAlert Mycoplasma Detection Kit (LONZA). Only mycoplasma negative cells were used. H1792, H2009 and A549 cell lines were exposed to increasing concentrations of Trametinib until they were completely resistant to the drug at the concentration of interest (0.5 μM) to obtain H1792-TR (Trametinib resistant), H2009-TR and A549-TR. H23 and H358 were treated with Sotorasib to generate resistant cells (H358-SR: resistant to 0,5 μM KRASi; H23: resistant to 10 μM KRASi). H2009, H23 and H358 cell lines were infected with pLENTI6-LacZ and pLENTI6-MYC as previously described[25].

## Reagents

Pharmacological inhibitors to MEK1/2 (Trametinib/GSK1120212, MEKi), HDAC (Panobinostat/LBH589, HDACi) and WEE1 (Adavosertib/MK-1775, WEE1i) were acquired from Selleckchem; to MEK5-K2 (BIX02189, MEK5i) was purchased from Tocris. Lestaurtinib/L6307 (mtPKCi), Neratinib (pan-HERi) and Afatinib (EGFRi) were obtained from LC Laboratories, and Midostaurin/PKC412 (mtPKCi), Sotorasib/AMG-510 (KRASi), Darovasertib (PKCAi), Barasertib (AURKBi), Adagrasib/

MRTX849 (KRASi) and MG132 (proteasome inhibitor) were purchased from MedChemExpress. Dabrafenib (RAFi) was a kind gift from Imanol Arozarena (Navarrabiomed, Spain).

## RNA sequencing

Samples were prepared with the Illumina TruSeq Stranded mRNA kit as per the manufacturer's indications and sequenced as reverse paired-end (100 bp) runs on the HiSeq 4000 sequencer. Raw fastq files were trimmed with Trimmomatic/0.36 and reads were aligned to the mm10 reference genome with STAR/2.5.1b aligner. Gene level counts were determined with STAR quantMode option using gene annotations from GENCODE (vM13). QC assessments such as unique alignment counts, unique/multiple ratio or exon/intron ratio was derived with ngsutilsj-0.3-2180ca6 using the bam-stats option. Differential gene expression and all other pathway analysis are conducted with R/3.4.3. Samples were imported, normalized with trimmed mean of Mvalues (TMM) from the EdgeR/3.20.9 package and further transformed with VOOM from the Limma/3.34.9 package, resulting in a log2 normalized count matrix. A linear model using the Limma/3.34.9 package was then used to obtain p-values, adjusted pvalues and log-fold changes (LogFC).

## Western blotting

Cells were scraped and lysed in buffer containing 1% NP-40, 150 mM NaCl, 50 mM Tris pH 7.4 and 1 mM EDTA, supplemented with protease inhibitor cocktail (Roche), 25 mM sodium fluoride, 1 mM sodium orthovanadate and 1.1 mM phenylmethylsulfonyl fluoride. Proteins were separated by SDS-PAGE electrophoresis in polyacrylamide gels with running buffer (0.25 M Tris, 1.92 M Glycine and 34.6 mM SDS) at 120 V, and then transferred to nitrocellulose membranes (BioRad) in ice-cold transfer buffer (0.21 M Tris and 1.92 M Glycine) at 110 V for 90–120 min. Membranes were blocked with 5% milk TBS-T for 1 h and then incubated overnight at 4 °C with primary antibody. Membranes were then incubated with secondary antibodies for 1 h at room temperature and developed using SuperSignal West Pico PLUS (Thermo Scientific) or Lumigen ECL Ultra TMA-6 (Lumigen) in Odyssey Fc Imager. Quantification of band intensity was done using ImageJ program (NIH-National Institutes of Health). Antibodies used: β-TUBULIN (1:2000, sc-9104, Santa Cruz Biotechnology), GAPDH (1:5000, ab9484, Abcam), ACTIN (1:5,000, A5441, Sigma), HSP90 (1:500, sc-69703, Santa Cruz Biotechnology), KRAS (1:1,000, WH0003845M1, Sigma), MYC (1:1,000, D84C12, #5605, Cell Signalling Technology-CST), phospho-MYC S62 (1:1000, ab185656, Abcam), phospho-MYC T48 (1:1000, ab28842, Abcam), ERK1/2 (1:1,000, #9102, CST), p-ERK1/2 (1:1000, #9101, CST), PARP (1:1000, #9542, CST), AKT (1:1000, #9272, CST), p-AKT (1:1000, #9271, CST), p70S6K (1:1000, #2708, CST), p-p70S6K (1:1000, #9205, CST), EGFR (1:1000, #2232, CST), p-EGFR (1:1000, #2236, CST), STAT3 (1:500, #4904, CST), p-STAT3 (1:1000, #9145, CST), cJUN (1:700, #9165, CST), SHP2 (1:1000, #3397, CST), phospho-SHP2 (1:1000, #3751, CST), FLT3 (1:500, #3462, CTS), and p-FLT3 (1:500, #3461, CTS).

## Drug combination studies in vitro

For 2D assays, cell lines were plated at density ranging from 1500 to 10,000 cells in 96-well plates and treated on the following day with single drugs or both. CellTiter 96® AQueous One Solution Cell Proliferation Assay (Promega) was used to determine the number of viable cells in proliferation and the potential cytotoxicity of drugs in cell lines. Experiments were read after 72 h of exposure to the drugs according to manufacturer's instructions. CompuSyn software (www.combosyn.com) was used to determine the potential synergism of two single drugs in the pairwise drug screen. Combination Index (CI) values lower than 0.8 were considered synergistic.

For 3D-cultures, cold 96-well plates were pre-treated with 50% Matrigel Growth Factor Reduced (MG) (Corning) coating before cell

seeding. Cells were resuspended in 10% Matrigel culture medium (DMEM F12, HEPES 1X, glutamax 1X, primocin 1X, 500 nM TGFβi/A83-01, 50 ng/mL mEGF, 100 ng/mL mNoggin, 100 ng/mL hFGF10, 0.01 µM gastrin I, 1.25 mM N-acetylcysteine, 10 mM nicotinamide, B-27 supplement 1X, R-spondin I-conditioned media 1X, Wnt3a-conditioned media 1X) and seeded at $2 \times 10^3$ cells per well. Cells were incubated overnight and drugs were added on the next day. Proliferation of 3D-cultures was measured using CellTiter-Glo® 3D Cell Viability Assay (Promega) according to manufacturer's instructions. Organoid images were taken using an inverted microscope DMI3000 from Leica.

## Clonogenic assay
Cells were seeded in triplicate into 24-well plates (range: 350–6000 cells per well depending on the cell line). The next day, cells were cultured in the absence or presence of single drugs or drug combinations for 10 days. Media with or without drugs was replaced every 3 days. Remaining cells were fixed with 4% formaldehyde (Panreac) for 15 min at RT, stained with crystal violet solution (Sigma-Aldrich) (1% crystal violet in $H_2O$) for 15 min and photographed using a digital scanner (EPSON Perfection v850 Pro). Relative growth was quantified by measuring absorbance at 570 nm in a spectrophotometer (SPEC-TROstar Nano – BMG Labtech) after extracting crystal violet from the stained cells using 20% of acetic acid (Sigma).

## Patient derived xenograft organoids
Organoids from TP60, TP79, TP80, TP181 and TP126 PDXs derived from LUAD samples were used. All these samples were whole-exome sequenced and confirmed to carry a *KRASG12C* mutation. Establishment and growth of PDXs was approved by the institutional Committee on Animal Research and Ethics of Hospital Virgen del Rocío, Hospital 12 Octubre, IBIS and CNIO under the protocol references SSA/SI/MD/pdm, PROEX 084/15 and PROEX 313/19, and patients consented for the use of their tissues to generate PDX models.

PDXs were minced into small fragments and enzymatically digested with collagenase in Basic medium (Advanced DMEM/F12, 1x HEPES, 1x Glutamine, 1X Primocin) at 37 °C for 60 min. After incubation, the digested samples were filtered using 70 µm filters and the disaggregated cells were centrifuged at $1500 \times g$ for 5 min. After two washes with Basic medium, the cells were counted and resuspended in Complete Feeding Media (Basic medium, 500 nM TGFβi/A83-01, 50 ng/mL mEGF, 100 ng/mL mNoggin, 100 ng/mL hFGF10, 0.01 µM gastrin I, 1.25 mM N-acetylcysteine, 10 mM nicotinamide, B-27 supplement 1X, R-spondin I-conditioned media 1X, Wnt3a-conditioned media 1X) with 10% of matrigel.

## Apoptosis assay
Alexa Fluor 647-conjugated Annexin-V (Invitrogen) was used to perform the apoptotic cell detection assays following the manufacturer's instructions. Cells were treated for 24 h, acquired in FACSCanto II Cytometer (BD Biosciences), and analysed using FlowJo software v9.3.

## Proteasome inhibitor assay
Two million of cells were plated in 10 cm² plates and treated on the following day with both drugs for 48 h and next, with 5 µM proteasome inhibitor MG132 for additional 6 h, before western blot analysis.

## Proteomics
Cellular pellets derived from H1792 cells exposed to DMSO, Trametinib (0.5 µM), Lestaurtinib (0.625 µM) or both for 48 h, and H1792-TR grown in 0.5 µM Trametinib ± Lestaurtinib for 48 h were subjected to proteomics using an isobaric tandem mass tag (TMT) approach LC-MS/MS analysis was done using a 5600 Triple-TOF system (Sciex). Cellular pellets were homogenized in lysis buffer containing 7 M urea, 2 M thiourea, 4% (w/v), and 50 mM DTT supplemented with protease and phosphatase inhibitors. The homogenates were spun down at

$100,000 \times g$ for 1 h at 15 °C. After protein precipitation, protein concentration was measured in the supernatants with the Bradford assay kit (Biorad).

– Protein digestion and peptide TMT labeling. Four independent tandem-mass-tag (TMT)-based quantitative proteomic experiments were performed including biological triplicates derived from biological conditions. TMT labeling of each sample was performed according to the manufacturer's protocol (Thermo). Briefly, equal amounts of protein (600 µg) from each sample were reduced with 200 mM tris (2-carboxyethyl) phosphine (TCEP) at 55 °C for 1 h. Cysteine residues were alkylated with 375 mM iodoacetamide at room temperature for 30 min. Protein enzymatic cleavage was carried out with trypsin (Promega; 1:40, w/w) at 37 °C for 16 h. Peptide desalting was performed using Pierce™ Peptide Desalting Spin Columns according to the manufacturer's instructions. For TMT experiments, each tryptic digest was labelled with one isobaric amine-reactive tag as follows: (i) TMT-Plex-1: Tag126, control-1; Tag127, control-2; Tag128, T-1; Tag129, T-2; Tag130, L-1; Tag131, L-2; (ii) TMT-Plex-2: Tag126, T-3; Tag127, M-3; Tag128, control-3; Tag129, control-4; Tag130, Comb-3; Tag131, TR-3; (iii) TMT-Plex-3: Tag126, TR-1; Tag127, TR-2; Tag128, Comb-1; Tag129, Comb-2; Tag130, control-5; Tag131, control-6; Tag126, control-7; Tag127, control-8; Tag128, TRL-1; Tag129, TRL-2; Tag130, TRL-3; Tag131, control-9. After 1 h incubation at room temperature, reactions were stopped with 5% hidroxilamine, labelled samples corresponding to the same plex were independently pooled, desalted and evaporated in a vacuum centrifuge.

– LC-MS/MS. Peptide pools were dried in a vacuum centrifuge and reconstituted with 40 µL of 5 mM ammonium bicarbonate (ABC) pH 9.8, and injected into an ÄKTA pure 25 system (GE Healthcare Life Sciences) with a high pH stable X-Terra RP18 column (C18; 2.1 mm x 150 mm; 3.5 µm) (Waters). Mobile phases were 5 mM ammonium formate in 90% ACN at pH 9.8 (buffer B) and 5 mM ammonium formate in water at pH 9.8 (buffer A). Column gradient was developed in an 80 min three step gradient from 5% B to 30% B in 5 min, 30% B to 60% B in 40 min, 15 min in 60% B and 60% B to 90% B in 20 min. Column was equilibrated in 95% B for 30 min and 2% B for 10 min. Thirty fractions were collected and evaporated under vacuum. Peptide fractions were reconstituted into a final concentration of 0.5 µg/µL of 2% ACN, 0.5% FA, 97.5% MilliQ-water prior to mass spectrometric analysis. Then, peptide mixtures were separated by reverse phase chromatography using an Eksigent nanoLC ultra 2D pump fitted with a 75 µm ID column (Eksigent 0.075 x 250). Samples were first loaded for desalting and concentration into a 2 cm length 100 µm ID precolumn packed with the same chemistry as the separating column. Mobile phases were 100% water 0.1% formic acid (FA) (buffer A) and 100% Acetonitrile 0.1% FA (buffer B). Non-modified peptide fractions were analyzed following the following conditions. Column gradient was developed in a 135 min three step gradient from 2% B to 30% B in 90 min, from 30% B to 40% B in 10 min and from 40% to 80% in 10 min. Column was equilibrated in 97% B for 3 min and 2% B for 23 min. During all the process, precolumn was in line with column and flow maintained all along the gradient at 300 nL/min. Eluting peptides from the column were analyzed using a 5600 Triple-TOF system (Sciex). Information data acquisition was acquired upon a survey scan performed in a mass range from 350 $m/z$ up to 1250 $m/z$ in a scan time of 250 ms. The top 35 peaks were selected for fragmentation. Minimum accumulation time for MS/MS was set at 11 ms giving a total cycle time of 3.8 s. Product ions were scanned in a mass range from 100 $m/z$ up to 1500 $m/z$ and excluded for further fragmentation for 15 s. In the case of fractions that contained the phosphorylated peptides, column gradient was developed in 140 min two-step gradient from 2% to 35% B in 100 min and from 35% to 70% in 20 min. Column was

equilibrated in 95% B for 5 min and 2% B for 15 min. Precolumn was in line with column and flow maintained all along the gradient at 300 nL/min. Eluting peptides from the column were analyzed using a 5600 Triple-TOF system (Sciex) following the same conditions as the non-modified peptides.

- Data analysis. Raw MS/MS spectra searches were processed using the MaxQuant software[68] and searched against the Uniprot proteome reference for Homo Sapiens (Proteome ID: UP000005640_9606, February 2019). The parameters used were as follows: Initial maximum precursor (25 ppm) fragment mass deviations (40 ppm); variable modification (methionine oxidation and N-terminal acetylation) and fixed modification (MMTS); enzyme (trypsin) with a maximum of 1 missed cleavage; minimum peptide length (7 amino acids); false discovery rate (FDR) for PSM and protein identification (1%). Frequently observed laboratory contaminants were removed. Protein identification was considered valid with at least one unique or "razor" peptide. The protein quantification was calculated using at least 2 razor + unique peptides, and statistical significance was calculated with a two-way Student-$t$ test ($p < 0.05$). A 1.2-fold change cut-off was used. Proteins with TMT ratios below the low range (0.8) were considered to be down-regulated, whereas those above the high range (1.2) were considered to be up-regulated. The Perseus software (version 1.5.6.0)[69] was used for statistical analysis and data visualization. The identification of significantly dysregulated regulatory/metabolic pathways across proteomic datasets was performed using Metascape[70].

## Preclinical in vivo models

All experiments in mice were performed following ARRIVE guidelines and approved by the institutional Committee on Animal Research and Ethics of CIMA and CNIO under the protocol numbers 057-18 and PROEX 316/19. Mice were kept in a 12:12 light/dark cycle with progressive increase or decrease of light intensity to mimic the dawn/twilight across the facility. The appropriate temperature (20–24 °C), humidity (50% +/− 10%) and pressure levels were provided and monitored daily. Mice were randomized to get a similar average tumor size across treatment groups at treatment start. Animals were treated without knowledge of anticipated outcomes and blind treatments were followed. The maximum tumor size authorized by the Committees on Animal Research and Ethics was 1000 mm$^3$ and was not exceeded at any time during the experiments.

For xenograft experiments $3 \times 10^6$ cells (H1792, A549, H1792-TR, H358-SR, H358-LacZ, H358-MYC and T1) were suspended in 100 μL of DPBS and injected subcutaneously into the two lower flanks of 10–12 weeks-old immune-deficient Rag2$^{-/-}$; Il2γr$^{-/-}$ mice (C- Rag2tm1Flv Il2rgtm1Flv; Jackson Laboratories) or immunocompetent F1 C57Bl/6 J x 129S2/Sv mice. C57Bl/6J and 129S2/Sv mice were purchased from Janvier. Beginning 1-week post-injection, tumor dimensions were measured every 3 days using a Digital caliper (DIN862, Ref 112-G, SESA Tools) and tumor volume was calculated by the formula: Volume = π/6 × length × width$^2$. When tumors reached an average volume of 100 mm$^3$, administration of pharmacological inhibitors was carried out: Trametinib (1 mg/kg), Lestaurtinib (30 mg/kg), Midostaurin (25 mg/kg), Sotorasib (10 or 30 mg/kg) or dual administration was done by oral gavage daily 5 days per week for 3 weeks. 10 to 12-week old KRas$^{FSFG12C}$; Trp53$^{FRT/FRT}$ mice[46] in a mixed C57Bl6J-129S4/Sv background were infected with AdFlp (10$^6$ p.f.u.) and aged for 6–8 months until mice had developed 1–4 tumors per lung. Then, tumors were treated with vehicle, Midostaurin (25 mg/kg), Sotorasib (100 mg/kg) or both by oral gavage for 6 weeks. Tumor follow up was done using microCT scans. In brief, lung images were acquired using SuperArgus COMPACT (Sedecal) microCT scanner. Image processing, analysis and 3D rendering was performed using the 3D Slicer Viewer Software.

## Immunohistochemistry

Immunohistochemical staining was performed using the EnVision TM + System (K400311-2, Agilent) according to the manufacture's recommendations. Antigen retrieval was performed for 30 min at 95 °C in Tris-EDTA, pH 9.0. The following rabbit primary antibodies were used: MYC (Abcam, ab32072) and CD8 (Cell Signalling, 98941).

## Real time PCR

RT-PCR was performed using SYBR® GreenER™Select Master Mix method (Applied Biosystems) on a QuantStudio 3 Real-Time PCR machine (ThermoScientific) following the manufacturer's instructions. *GAPDH* was used as a housekeeping gene. Primers used for RT-PCR were as follows: 'MYC forward' is 5′-GCTGCTTAGACGCTGGATTT-3′, 'MYC reverse' is 5′-TAACGTTGAGGGGCATCG-3′, 'GAPDH forward' is 5′-GAGTCAACGGATTTGGTCGT-3′ and 'GAPDH reverse' is 5′-AAGT-GAAGGGGTCATTGATGG-3′.

## Lentiviral infections

A *MYC* cDNA in a pDONR221 vector was provided by Alejandro Sweet-Cordero (University of California San Francisco, USA) and cloned into a pLenti6/V5-DEST using the Gateway system (Thermofisher). Lentivirus were produced by transfection of 2 μg of lentiviral plasmid, 0.5 μg of packaging plasmid (psPAX2 – Addgene; #12260) and 0.7 μg of envelope plasmid (pMD.G2 – Addgene; #12259) into HEK293T cells using X-tremeGENE HP DNA Transfection Reagent (Roche). Forty-eight hours later, supernatant was harvested, filtered and applied directly to cells for infection at a MOI lower than 1.

## Statistics and reproducibility

Statistical analyses were performed using GraphPad Prism 8 (Graph-Pad Software, Inc). For in vitro experiments, at least 3 independent experiments with similar results were carried out with 2–6 replicates per experiment. Sample size was chosen using http://www.biomath.info/power/ttest.htm or based on similar experiments previously published by the authors. For comparisons of two groups, samples were explored for normality (Shapiro-Wilk test) and variance (Levene test). Groups with normal distribution of samples followed a $t$-test. Non-normal samples were analysed using a Mann–Whitney test (equal variances) or a Median test (unequal variances). All analyses were two-tailed. For multiple comparisons of normally distributed variables, ANOVA and posterior Tukey tests were carried out. In the case of multiple comparisons of non-normally distribution variables, Kruskal–Wallis and posterior Tukey Adjusted-Mann–Whitney $U$ tests were used. Statistical significance was defined as significant ($p < 0.05$), very significant ($p < 0.01$) and highly significant ($p < 0.001$). Error bars correspond to either standard deviation ($n < 8$) or standard error of the mean ($n \geq 8$), as indicated for each experiment. No data points were removed as outliers.

## Reporting summary

Further information on research design is available in the Nature Portfolio Reporting Summary linked to this article.

## Data availability

The datasets generated during and/or analysed during the current study are available in public repositories: RNAseq files can be found at GSE161218 [https://www.ncbi.nlm.nih.gov/geo/query/acc.cgi?acc=GSE161218]; search results files and MS raw data of proteomics analyses were deposited in the ProteomeXchange Consortium (http://proteomecentral.proteomexchange.org) via the PRIDE partner repository with the dataset identifiers PXD024023 (Project Webpage: http://www.ebi.ac.uk/pride/archive/projects/PXD024023; FTP Download: ftp://ftp.pride.ebi.ac.uk/pride/data/archive/2023/08/PXD024023]. Publicly available data sets used in this study are referenced in the text and figure legends[26–31]. The publicly available data used in this

study from Wilkerson et al.[26], Okayama et al.[27], Ding et al.[29], and Beer et al.[30] are available in the GEO database under accession codes: GSE36471; GSE31210; GSE12667; GSE68571. The raw data (Affymetrix CEL files) from the Chitale et al. microarray sample set[31] is provided online at http://cbio.mskcc.org/Public/lung_array_data/. All of the primary sequence files from Collisson et al.[28] are deposited in cgHub and all other data are deposited at the Data Coordinating Center (DCC) for public access (http://cancergenome.nih.gov/), (https://cghub.ucsc.edu/) and (https://tcga-data.nci.nih.gov/docs/publications/luad_2014/). Source data are provided with this paper. The remaining data are available within the Article, Supplementary Information or Source Data file. Biological material (e.g. cell lines) generated in this study is available on the basis of a Material Transfer Agreement. Additional reagents will be made available upon reasonable request. Source data are provided with this paper.

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

## Acknowledgements

We thank members of the Solid Tumors Program at CIMA for insightful comments and the staff at the Flow Cytometry, Animal, and Genomics facilities for technical assistance. M.R. was supported by a fellowship from MICIU (FPU15/00173), R.E. by a donation from Mauge Burgos de la Iglesia's family. The Proteomics Platforms of Navarrabiomed led by E.S. and J.F.-I. is a member of Proteored, PRB3 and is supported by grant PT17/0019 of the PE I + D + i 2013–2016, funded by ISCIII and ERDF. E.J.-L. was supported by Foundation of Spanish Association Against Cancer (PROYE18012ROSE), by Centro de Investigación Biomédica en Red (CIBERONC; CB16-12-00350), and by Generalitat Valenciana (AICO/2021/333). F.L. was funded by the Gobierno de Navarra (Ref. 34/2021), the Cancer Research Thematic Network of the Instituto de Salud Carlos III (RTICC RD12/0036/0066), PID2021-122638OB-I00 MCIN/AEI/10.13039/501100011033/ FEDER, UE and by FEDER "Una manera de hacer Europa". I.F. was funded by FIS PI19/00320 and by the Miguel Servet Program CP21/00052. S.V. was supported by Ministerio de Ciencia, Innovación y Universidades, Convocatoria 2019 para incentivar la Incorporación Estable de Doctores (IED2019-001007-I), by FEDER /Ministerio de Ciencia, Innovación y Universidades - Agencia Estatal de Investigación (SAF2017-89944-R and PID2020-116344-RB-100/MCIN/AEI/10.13039/501100011033, by a Leonardo Grant for Researchers and Cultural Creators 2018 from BBVA Foundation, by a seed grant at the I Convocatoria Proyectos Prueba de Concepto from PRB3-Proteored (Institute of Health Carlos III-ISCIII), by Fundació La Marató de TV3 (474/C/2019), and by Foundation of Spanish Association Against Cancer - Strategic Projects 2020 (PROYE20029VICE). M.P.-S. and S.V. were also funded by Fundación Alberto Palatchi. None of the funding sources were involved in the decision to submit the article for publication.

## Author contributions

I.M., M.R. and S.V. conceived and designed the project. S.V. supervised the work. I.M., M.R., E.S., J.F.-I., I. Ferrer, L.P.-A., M.D., M.B., and S.V. designed and planned the experiments. I.M., M.R., C.W., R.E-C., M.S., A.S., I. Feliu, J.K., I.L., M.R.-R., M.F., S.C., D.L.-A., E.J.-L., S.N., M.P.-S and F.L. did the experiments. S.L. and P.K. performed repurposing analysis. E.G. carried out RNAseq and GSEA computational work, and survival analyses. I.A.L. and A.P.-L. contributed reagents. S.P.-E., E.S. and J.F.-I. performed proteomics studies and related computational analysis. I.M., M.R., D.R., E.J.-L., E.S., J.F.-I., I.F., L.P.-A., J.F.-I., M.D., M.B., and S.V. were responsible for the data analysis and interpretation. I.M., M.R. and S.V. wrote the manuscript and were in charge of the manuscript

preparation. All the authors reviewed, edited, and approved the manuscript.

## Competing interests

M.P.-S. reports personal fees from Incyte, Genentech, Hoffmann La Roche, TFS Health Science and Astra Zeneca; research funding from Roche and BMS; and travel grants from Incyte and BMS. F.L. and S.V. report research funding from Roche. S.V. discloses project funding from Revolution Medicines and advisor fees from Libera Bio. None of the disclosed information applies to the current project. No potential conflicts of interest were disclosed by the other authors.

## Additional information

Irati Macaya [1,26], Marta Roman[1,23,26], Connor Welch[1], Rodrigo Entrialgo-Cadierno [1], Marina Salmon[2,3], Alba Santos [3,4], Iker Feliu[1], Joanna Kovalski [5,6], Ines Lopez[1], Maria Rodriguez-Remirez[1], Sara Palomino-Echeverria[7], Shane M. Lonfgren[8,9], Macarena Ferrero [3,10,11], Silvia Calabuig[3,10,11,12], Iziar A. Ludwig [13], David Lara-Astiaso[14], Eloisa Jantus-Lewintre [3,10,11,12], Elizabeth Guruceaga [15,16,17], Shruthi Narayanan[1,18], Mariano Ponz-Sarvise [1,16,18], Antonio Pineda-Lucena[13], Fernando Lecanda[1,3,16,19], Davide Ruggero [5,6,20], Purvesh Khatri [6,7], Enrique Santamaria [16,17], Joaquin Fernandez-Irigoyen [16,17], Irene Ferrer[3,4], Luis Paz-Ares[3,4,21,22], Matthias Drosten [2,24], Mariano Barbacid[2,3], Ignacio Gil-Bazo [1,3,16,18,25,27] & Silve Vicent [1,3,16,19,27] ✉

¹University of Navarra, Center for Applied Medical Research, Program in Solid Tumors, Pamplona, Spain. ²Experimental Oncology Group, Molecular Oncology Program, Spanish National Cancer Center (CNIO), Madrid, Spain. ³Centro de Investigación Biomédica en Red de Cáncer (CIBERONC), Madrid, Spain. ⁴H12O-CNIO Lung Cancer Clinical Research Unit, Instituto de Investigación Hospital 12 de Octubre & Centro Nacional de Investigaciones Oncológicas (CNIO), Madrid, Spain. ⁵Helen Diller Family Comprehensive Cancer Center, University of California San Francisco, San Francisco, CA, USA. ⁶Department of Urology, University of California San Francisco, San Francisco, CA, USA. ⁷Navarrabiomed, Complejo Hospitalario de Navarra (CHN), Universidad Pública de Navarra, Pamplona, Spain. ⁸Stanford Institute for Immunity, Transplantation and Infection, Stanford, CA, USA. ⁹Stanford Center for Biomedical Informatics Research, Department of Medicine, Stanford University, Stanford, CA, USA. ¹⁰Molecular Oncology Laboratory, Fundación Para La Investigación del Hospital General Universitario de Valencia, Valencia, Spain. ¹¹Mixed Unit TRIAL (Principe Felipe Research Centre & Fundación para la Investigación del Hospital General Universitario de Valencia), Valencia, Spain. ¹²Department of Pathology, Universitat de Valencia, Valencia, Spain. ¹³University of Navarra, Center for Applied Medical Research, Molecular Therapies Program, Pamplona, Spain. ¹⁴University of Navarra, Center for Applied Medical Research, Genomics Platform, Pamplona, Spain. ¹⁵University of Navarra, Center for Applied Medical Research, Bioinformatics Platform, Pamplona, Spain. ¹⁶IdiSNA, Navarra Institute for Health Research, Pamplona, Spain. ¹⁷ProteoRed-Instituto de Salud Carlos III (ISCIII), Madrid, Spain. ¹⁸Clinica Universidad de Navarra, Department of Medical Oncology, Pamplona, Spain. ¹⁹University of Navarra, Department of Pathology, Anatomy and Physiology, Pamplona, Spain. ²⁰Department of Cellular and Molecular Pharmacology, University of California San Francisco, San Francisco, CA, USA. ²¹Medical Oncology Department, Hospital Universitario 12 de Octubre, Madrid, Spain. ²²Medical School, Universidad Complutense, Madrid, Spain. ²³Present address: Division of Hematology and Oncology, University of California San Francisco, San Francisco, CA, USA. ²⁴Present address: Molecular Mechanisms of Cancer Program, Centro de Investigación del Cáncer and Instituto de Biología Molecular y Celular del Cáncer, CSIC-University of Salamanca, Salamanca, Spain. ²⁵Present address: Department of Oncology, Fundación Instituto Valenciano de Oncología, Valencia, Spain. ²⁶These authors contributed equally: Irati Macaya, Marta Roman. ²⁷These authors jointly supervised this work: Ignacio Gil-Bazo, Silve Vicent. ✉e-mail: silvevicent@unav.es

