## [Peer Review File · Nature Communications]

Reviewers' Comments:

Reviewer #1:

Remarks to the Author:

The authors have addressed the initial concerns, and as such the revised manuscript is significantly strengthened.

Reviewer #2:

Remarks to the Author:

The authors met this reviewer's expectations revising the manuscript.

Reviewer #3:

Remarks to the Author:

The authors have improved the study by starting to identify the targets of midostaurin that might be mediating the enhanced toxicity they see in their lung cancer models. Several candidates seem to have been identified, which is a good start. It is possible, of course, that there are many midostaurin targets that are relevant. I would have more enthusiasm if the authors could be more convincing that the ones they identify are truly important. It would have been ideal to knockdown each of the known midostaurin targets and ask which ones have some additive effect with trametinib. The value of knowing the mechanism is that polykinase inhibitors, like midostaurin, have significant side effects in patients, and one might reasonably therefore like to use a more specific drug with hopefully less side effects.

Please find on the next pages a point-by-point response the comments raised by reviewer 2 with regard to our manuscript entitled “SIGNATURE-DRIVEN REPURPOSING OF MIDOSTAURIN FOR COMBINATION WITH MEK1/2 AND KRASG12C INHIBITORS IN LUNG CANCER”. I would like to thank you the reviewers review for helping increase the value of this work.

REVIEWERS' COMMENTS

Reviewer #1 (Remarks to the Author):

The authors have addressed the initial concerns, and as such the revised manuscript is significantly strengthened.

We thank the reviewer for acknowledging the work aimed to respond to the original concerns.

Reviewer #2 (Remarks to the Author):

The authors have improved the study by starting to identify the targets of midostaurin that might be mediating the enhanced toxicity they see in their lung cancer models. Several candidates seem to have been identified, which is a good start. It is possible, of course, that there are many midostaurin targets that are relevant. I would have more enthusiasm if the authors could be more convincing that the ones they identify are truly important. It would have been ideal to knockdown each of the known midostaurin targets and ask which ones have some additive effect with trametinib. The value of knowing the mechanism is that polykinase inhibitors, like midostaurin, have significant side effects in patients, and one might reasonably therefore like to use a more specific drug with hopefully less side effects.

We thank the reviewer for stressing the importance of identifying the midostaurin targets involved in the sensitization to Trametinib (and Sotorasib).

In response to his/her concern, for the last three months we have attempted to inhibit PRKCA and AURKB using two different strategies based on lentivectors expressing shRNA and sgRNAs (pLKO.1-puro and pLentiCRIPRv2-puro respectively). However, the generation of pooled knockdown cells from H23 and H358 cells has revealed that AURKB abrogation leads to a strong antiproliferative/cytotoxic phenotype in the tumor cells (see image below). In tune with this observation, we have been unable to obtain single AURKB-knockout CRISPRed clones. Collectively, this data suggests a strong dependence of mutant KRAS lung cancer cells on this gene, what has precluded the combinatorial studies with Trametinib and Sotorasib with this lentivectors.

In order to circumvent this technical limitation and also to use the same delivery vector to deplete AURKB and PRKCA, we have used a doxycycline inducible vector which allowed us to deplete gene expression levels at the time of treatment administration (once the cells had been puromycin selected in the absence of shRNA induction). As shown in the figure below, we managed to obtain pooled populations with efficient knockdown of AURKB and PRKCA expression (A). Related to AURKB, although we still see that single AURKB inhibition leads to a strong antiproliferative effect in these cells (>50% proliferation decrease), these conditions allow to see some additive antiproliferative effect when Trametinib or Sotorasib are added (B). With regard to PRKCA, we do not observe that its depletion sensitizes to Trametinib or Sotorasib (C). The discrepancies between the pharmacological and genetic data for PRKCA could be explained by the fact that there is no PRKCA specific inhibitor and the one used (Darovasitib) while being more active against PRKCA, it also targets other kinases such as PRKCθ, or GSK3β which could be mediating the sensitization to Trametinib and Sotorasib. For this reason, we have removed the PRKCAi-based combinations from the manuscript.

In summary, on the one hand, the data support the participation of AURKB as a Midostaurin target mediating the effect to this multikinase inhibitor. On the other hand, these new observations highlight the relevance of our pharmacological combination studies showing a synergistic effect between an AURKBi in combination with MEK1/2 or RASG12Ci. The pharmacological approach allows for the use of drug concentration lower than the maximum effective dose which are unlikely to completely inactivate each kinase (in our particular studies, such concentrations are lower than the IC50 for both drugs). We believe that having found a synergistic and robust antiproliferative effect with sub-optimal concentrations of the tested drugs (including combinations of Midostaurin with both Trametinib and Sotorasib) is a relevant aspect of our study, as one could expect that these conditions would open the room to also use lower drug concentrations in patients and, thus, decrease the overall toxicity.

Additionally, to complement the finding that AURKB is a relevant target of Midostaurin in mutant KRAS lung cancer, we present data derived from another project in the lab aimed to identify drugs that increase T cell cytotoxic activity against lung cancer. In this study, we capitalized on a co-culture assay consisting of mouse cells derived from a *KRAS*^{LSLG12D} lung cancer GEM model expressing the OVA antigen (KLA-OVA) and CD8+ T lymphocytes from OT.1 mice (which recognize mouse cells expressing the OVA antigen). Pharmacological screening of ~300 FDA-approved drugs unveiled Midostaurin and the Barasertib among the top 10 drugs enhancing the cytotoxic activity of CD8+ T cells. Collectively, these results support the idea that AURKB is a functional Midostaurin target in mutant KRAS-driven lung cancer.

Reviewer #3 (Remarks to the Author):

The authors met this reviewer's expectations revising the manuscript

We are grateful to the viewer for his/her positive comments.

Reviewers' Comments:

Reviewer #2:

Remarks to the Author:

I reviewed the prior revision of this manuscript and indicated that the reviewers had responded adequately to all concerns.

Point-by-point response to the comments raised by reviewer 2 with regard to our manuscript entitled "SIGNATURE-DRIVEN REPURPOSING OF MIDOSTAURIN FOR COMBINATION WITH MEK1/2 AND KRASG12C INHIBITORS IN LUNG CANCER".

REVIEWERS' COMMENTS

Reviewer #2 (Remarks to the Author):

I reviewed the prior revision of this manuscript and indicated that the reviewers had responded adequately to all concerns.

We thank the reviewer for his/her positive comments to the newly generated data.